# Binding of heterochromatin protein Rhino to a subset of piRNA clusters depends on a combination of two histone marks

Abdou Akkouche[1,3], Emma Kneuss [2,3], Susanne Bornelöv [2,3], Yoan Renaud [1], Evelyn L. Eastwood[2], Jasper van Lopik[2], Nathalie Gueguen[1], Mingxuan Jiang [2], Pau Creixell[2], Stéphanie Maupetit-Mehouas[1], Anna Sobieszek [2], Yifan Gui [2], Benjamin Czech Nicholson [2] ✉, Emilie Brasset [1] ✉ & Gregory J. Hannon [2] ✉

Animal germ cells deploy a specialized small RNA-based silencing system, called the PIWI-interacting RNA (piRNA) pathway, to prevent unwanted expression of transposable elements (TEs) and maintain genome integrity. In *Drosophila melanogaster* germ cells, the majority of piRNA populations originate from dual-strand piRNA clusters, genomic regions highly enriched in TE fragments, via an elaborate machinery centered on the Heterochromatin Protein 1 homolog, Rhino. Although Rhino binds to peptides carrying tri-methylated H3K9 in vitro, it is not fully understood why in vivo only a fraction of H3K9me3-decorated heterochromatin is occupied by Rhino. Recent work revealed that Rhino is recruited to a subset of piRNA clusters by Kipferl. Here we identify a Kipferl-independent mode of Rhino recruitment that, in addition to the previously established role of H3K9me3, also depends on the histone H3 lysine 27 methyltransferase Enhancer of Zeste. At Kipferl-independent sites, we find that Rhino specifically binds to loci marked by both H3K9me3 and H3K27me3 via its chromodomain. Although the exact mechanism of how Rhino binding is influenced by dual histone modifications remains unclear from a structural and biochemical perspective, our work suggests that combinatorial modifications may regulate the specificity of chromatin-binding protein interactions. These findings provide an enhanced understanding of how Rhino targets piRNA source loci, highlighting the sophisticated epigenetic landscape governing TE silencing in *Drosophila* germ cells.

Transposable elements (TEs) are genetic sequences, present in nearly all organisms, that are able to move and insert themselves into different positions within a genome. While TEs can contribute to genetic diversity and evolution, their uncontrolled activity in gonadal cells can have detrimental effects, including the potential to dramatically reduce fertility. In animal gonads, a mechanism known as the PIWI-interacting RNA (piRNA) pathway plays a crucial role in silencing and controlling active TEs to safeguard germline integrity[1–3]. The piRNA machinery utilizes 23–30-nt small RNAs, which associate with PIWI-family proteins, directing recognition of target TEs for post-transcriptional degradation or cotranscriptional silencing. While some piRNAs are derived from active TEs, the majority originate from dedicated piRNA source loci (piSL). Of those, a subset, called piRNA clusters, are enriched in TE content. All reported piRNA clusters in somatic follicle cells resemble canonical genes, give rise to piRNAs from one genomic strand and are referred to as unistrand clusters[4]. In contrast, piRNA clusters in the germline

are typically expressed non-canonically, producing piRNAs from both genomic strands and are referred to as dual-strand clusters[5–10].

Rhino (Rhi), a homolog of Heterochromatin Protein 1a (HP1a), is a germline-specific piRNA pathway factor essential for piRNA production, TE repression and fertility[9]. Rhi and HP1a belong to a distinct group of chromatin-binding proteins containing chromodomains (CDs), which display specific binding properties. For instance, *Drosophila* HP1a is associated with H3K9me2/3-enriched regions such as constitutive heterochromatin, whereas Polycomb (Pc), another CD protein, specifically binds to H3K27me3 often found in facultative heterochromatin[11–13]. In line with Rhi being evolutionary related to HP1a, several studies have shown that in vivo Rhi specifically localizes to a subset of piSL, including dual-strand piRNA clusters, which are typically characterized by the presence of H3K9me3 marks[6,8,9,14]. In vitro, the Rhi CD was shown to recognize di-/tri-methylated H3K9 peptides[6,14,15].

At the genomic regions to which it binds, Rhi acts as nucleator for a multiprotein complex that triggers non-canonical transcription from heterochromatic regions and facilitates the export of the resulting piRNA precursors from the nucleus[6–10,14,16–21]. Although Rhi binds H3K9me3 in vitro[6,14,15], Rhi immunofluorescence staining and previously published chromatin immunoprecipitation followed by sequencing (ChIP–seq) data indicate that in vivo Rhi associates only with a fraction of genomic regions enriched in H3K9me3 (ref. 6), hence raising questions about how its specificity is determined in vivo. A recent study identified the zinc finger domain protein Kipferl (Kipf) as a Rhi co-factor, with their interaction mediated by the CD of Rhi[22,23]. Kipf recruits Rhi to dedicated heterochromatic loci that exhibit enrichment in guanosine-rich motifs, suggesting that sequence content within piSL could contribute to the binding specificity of Rhi. However, while the interaction between Kipf and Rhi is critical for the recruitment of Rhi to numerous genomic loci, prominent Rhi-dependent piSL, such as piRNA clusters *42AB* and *38C*, show little to no dependence on Kipf[22]. These observations suggest that additional cues contribute to Rhi binding. Here, we uncover an unexpected, Kipf-independent mechanism for Rhi targeting that relies on the H3K27me3 methyltransferase Enhancer of Zeste (E(z)). We propose a model in which the combined recognition of H3K9me3 and H3K27me3 guides Rhino to a subset of piSL, including the most prominent cluster *42AB*.

## E(z) is required for TE silencing in *Drosophila* germ cells

Understanding how Rhi is recruited to and interacts with chromatin at piSL is crucial for unraveling the complexities of TE silencing and genome integrity. We therefore searched for factors affecting TE control in germ cells through a focused reverse-genetic screen in *Drosophila melanogaster* ovaries (Fig. 1a). We specifically targeted chromatin proteins and histone modifiers and probed the effects of their depletion on germline-specific TEs by quantitative RT–PCR (RT–qPCR). Knockdown of 32 genes resulted in TE derepression and highlighted potential roles of different protein complexes in TE regulation (for example, COMPASS complex and NSL complex) (Extended Data Fig. 1).

As expected from previous reports[24], depletion of the H3K9 methyltransferase Eggless (Egg) resulted in a strong deregulation of germline TEs and a loss of Rhi foci by immunofluorescence (Extended Data Figs. 1 and 2a). Unexpectedly, however, depletion of the histone methyltransferase E(z) resulted in TE upregulation (Extended Data Fig. 1). Of note, we also found that the depletion of other subunits of the Polycomb repressive complex 2 (PRC2), namely Su(z)12 and Caf1-55, resulted in the derepression of some TEs (Supplementary Note 1). In *Drosophila*, E(z) is responsible for H3K27me3 and is essential for Polycomb-mediated chromatin silencing[25,26]. Depletion of E(z) using a strong Gal4 driver (UAS-Dcr2 transgene combined with nos-Gal4) caused rudimentary ovaries. To exclude the possibility that the observed TE derepression was due to pleiotropic oogenesis defects, we used different Gal4 drivers, namely nos-Gal4, which is active from

the germarium through stage 2, then inactive between stages 3 and 6 of oogenesis but reactivated at later stages[27], and pTOsk-Gal4, which is expressed immediately after germline cyst formation[28], and two independent E(z) RNAi lines, all of which resulted in nearly wild-type ovarian morphology despite reduced H3K27me3 levels in nurse cell nuclei (Extended Data Fig. 2b), and unchanged Rhi localization as observed by immunofluorescence (Extended Data Fig. 2a). Sequencing of polyA-selected RNAs from ovaries depleted of E(z) using the nos-Gal4 driver revealed a strong upregulation of 13 germline TE families, including *gypsy12*, *burdock* and *copia* (Fig. 1b). This was validated by RNA-fluorescence in situ hybridization (FISH), with *burdock*, *gypsy12* and *copia* expression detected in germ cells depleted of E(z) (Fig. 1c). Overall, however, the effects on TEs that we observed for *E(z)* knockdown were less severe compared with ovaries depleted of Rhi, which affected 53 TE families, suggesting partial contributions of E(z) to Rhi function (Extended Data Fig. 2c; data from ref. 22).

To distinguish between compromised global chromatin silencing and a specific role in the piRNA pathway, we performed RNA-FISH for precursor transcripts derived from the dual-strand piRNA clusters *42AB* (Kipf independent) and *80F* (Kipf dependent), and the unistrand cluster *20A* (germline enriched but Rhi independent). Knockdown of *E(z)* revealed a strong loss in piRNA precursors derived from *42AB*, but not from *80F* and *20A*, suggesting a potential role of E(z) in transcription of Kipf-independent dual-strand piRNA clusters (Fig. 1d and Extended Data Fig. 2d).

Next, we analyzed the effects of *E(z)* knockdown on piRNA precursors globally using ribo-depleted RNA-sequencing (RNA-seq). To measure transcription across all piSL, we divided the genome into 1 kb bins and quantified the signal per bin, as previously described[6]. Upon germline-specific E(z) depletion, we observed a strong reduction in precursor transcripts across *42AB*, and moderately reduced levels for *38C*, whereas *20A* or *flam* (soma-expressed and Rhi-independent) were essentially unchanged (Fig. 1e and Extended Data Fig. 2e). Strikingly, precursor levels for Kipf-dependent piRNA cluster *80F* were increased. RT–qPCR confirmed the RNA-seq results (Extended Data Fig. 2f). Of note, and as reported previously[6,9,14], Rhi-depleted ovaries showed a specific and global reduction of precursor transcripts from dual-strand clusters (Extended Data Fig. 2g). These data suggest a more nuanced role of E(z) in TE silencing compared with Rhi.

Next, we sequenced piRNAs from ovaries in which E(z) or Rhi were depleted specifically in germ cells and compared these with control knockdowns (Supplementary Table1). Consistent with the reduction of precursor transcripts from *38C* and *42AB* upon *E(z)* knockdown (Fig. 1e), we detected a severe reduction in piRNA levels (Fig. 1f). As expected from their precursor expression, piRNAs from *80F* increased, while piRNAs from the unistrand clusters *20A* and *flam* were unaffected (Fig. 1f and Extended Data Fig. 2h). The reduction in precursor and piRNA levels observed at cluster *42AB* was similar for ovaries depleted for E(z) and Rhi (Fig. 1g). In contrast, precursors and piRNAs were increased at *80F*, whereas there was a pronounced reduction of these RNA populations in the *rhi* knockdown (Fig. 1h).

E(z)-depleted ovaries also showed fewer antisense piRNAs targeting individual germline-specific TEs, which could explain their increased mRNA levels (Extended Data Fig. 2i). As expected, piRNAs against TEs that are mainly active in the somatic compartment of the ovary (for example, *Tabor*, *gypsy* and *ZAM*), were not affected, probably due to their origin from the soma-specific *flam* cluster (Extended Data Fig. 2i,j). Altogether, our results reveal that the histone methyltransferase E(z) is required for TE silencing and that its depletion affects piRNA production in germ cells from several dual-strand clusters.

## H3K27me3 and H3K9me3 marks coexist on E(z)-dependent piSL

To elucidate why certain piSL require E(z), we examined the genome-wide distribution of H3K27me3, which is deposited by E(z),

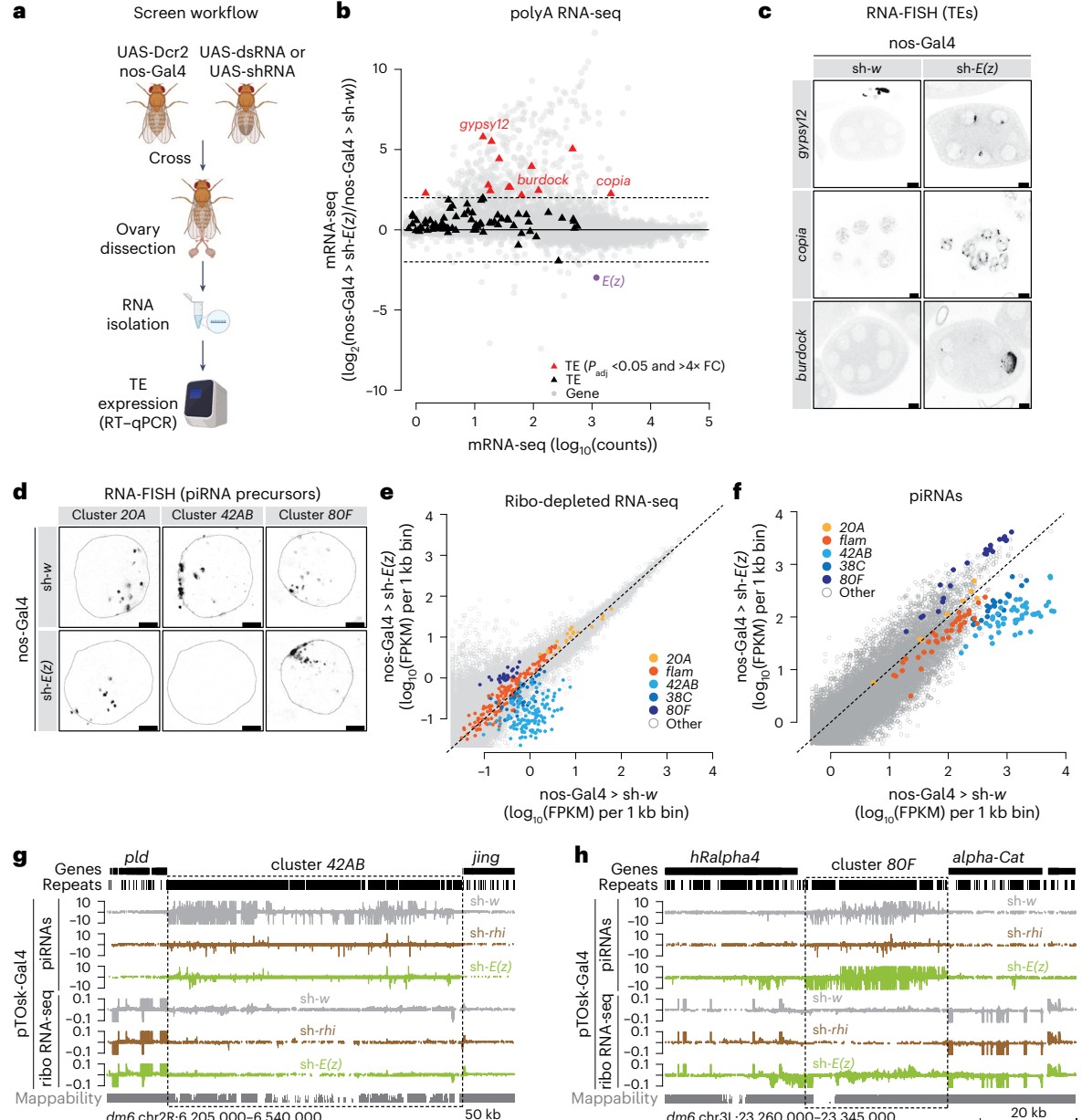

**Fig. 1 | E(z) is required for TE silencing in *Drosophila* germ cells and affects piRNA production. a**, A schematic showing the in vivo screen workflow. Male flies carrying dsRNA or shRNA constructs under the inducible UAS promoter were crossed to females expressing the germline-specific nos-GAL4 driver and a *Dcr2* transgene. F1 offspring were transferred to new vials and fertility was determined by counting eggs laid and hatched. RNA was extracted from ovaries of F1 offspring and TE levels were measured by RT–qPCR. **b**, An MA plot showing counts per gene (gray) and TEs (black) in polyA-selected RNA-seq libraries from ovaries depleted of w (control) and E(z) using the nos-Gal4 driver. TEs with *P*adj <0.05 and >4× fold change (FC) are shown in red, *E(z)* is indicated in purple. **c**, Confocal images showing RNA-FISH signal for the indicated TEs (*burdock*, *copia* and *gypsy12*) upon nos-Gal4 mediated knockdown or *w* or *E(z)*. Scale bar, 10 μm. Data are representative of *n* ≥ 3 independent experiments. **d**, Confocal images showing RNA-FISH signal for transcripts derived from piRNA clusters *42AB, 80F*

and *20A* in control and E(z)-depleted ovaries using the nos-Gal4 driver. Scale bar, 5 μm. Data are representative of *n* ≥ 3 independent experiments. **e**, A scatter plot depicting normalized ribo-depleted RNA levels (FPKM) of uniquely mapping reads in 1 kb bins in ovaries with nos-Gal4-driven *E(z)* knockdown versus control (average of three replicate experiments each). **f**, A scatter plot depicting normalized piRNA levels of uniquely mapping piRNAs (FPKM) in 1 kb bins (118,294 in total) in ovaries with nos-Gal4-driven *E(z)* knockdown versus control (two replicates each). **g**, UCSC genome browser tracks displaying the dual-strand piRNA cluster *42AB*. Levels of uniquely mapping piRNAs and ribo-depleted RNAs from ovaries with the indicated knockdowns using the pTOsk-Gal4 driver (average of three replicate experiments each) are shown along with tracks displaying genes, repeats and mappability. The dashed line indicates approximate piRNA cluster boundaries. **h**, As in **g** but showing the dual-strand piRNA cluster *80F*.

and H3K9me3, which has been associated with piRNA clusters and Rhi recruitment, by performing ChIP–seq and CUT&RUN (Extended Data Fig. 3a). We found that a number of genomic regions, including dual-strand piRNA cluster *42AB*, are decorated by both H3K9me3 and H3K27me3 (Fig. 2a,b). A similar, albeit less strong, pattern was observed at other piSL, including the *ey/Sox102F* region on chromosome 4

(Extended Data Fig. 3b). Notably, Rhi preferentially occupied regions decorated both by H3K9me3 and H3K27me3 over those only carrying H3K9me3 (Fig. 2c). Additionally, the majority of H3K27me3 peaks (76%) did not overlap with Kipf (Extended Data Fig. 3c). To determine whether one of the two marks, or both, could best predict piRNA clusters, we divided the genome into 1 kb bins and ranked these based on their

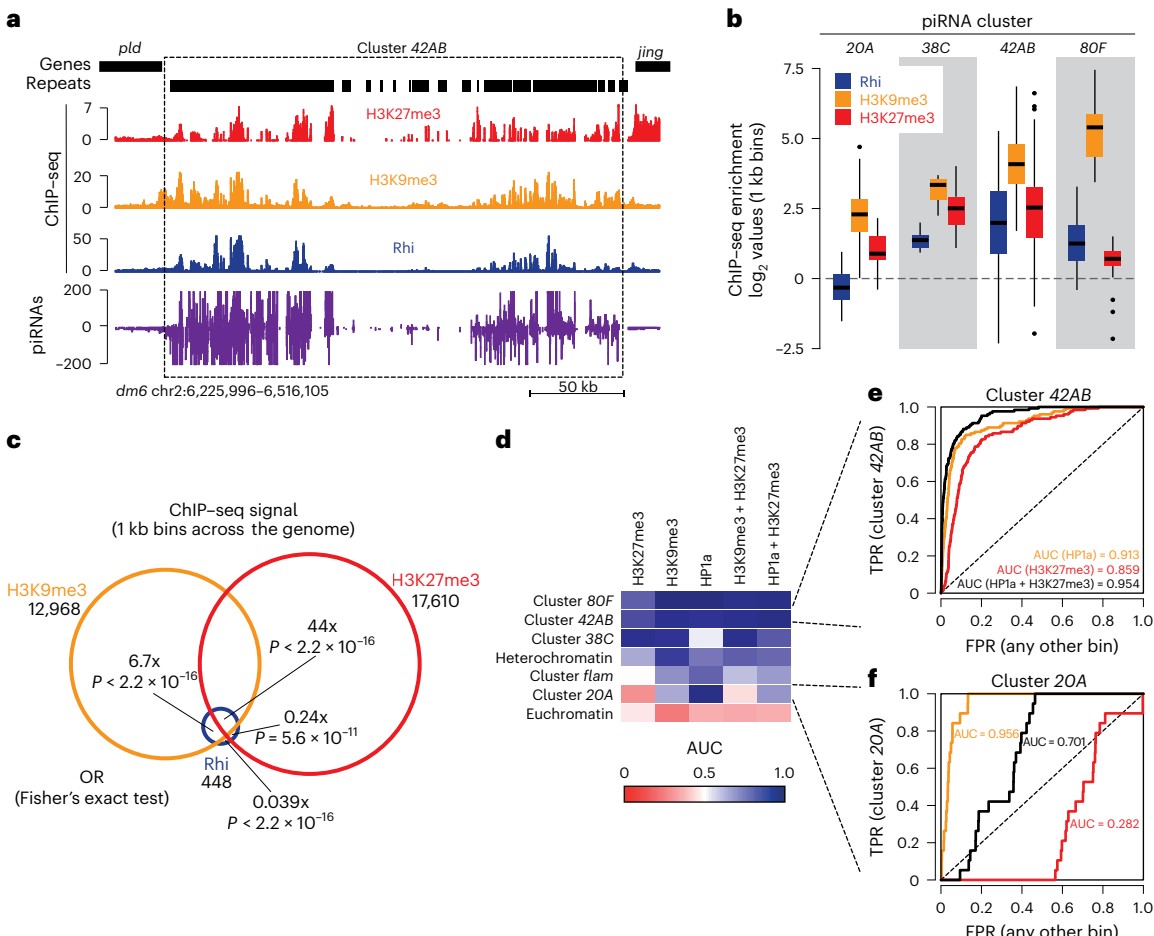

**Fig. 2 | H3K27me3 is present on certain piSL. a**, UCSC genome browser tracks of the region comprising dual-strand piRNA cluster *42AB* displaying ChIP–seq signal (depicted as coverage per million reads) of Rhi (blue), H3K27me3 (red), H3K9me3 (orange), uniquely mapping piRNAs (purple, coverage normalized to miRNA reads) and tracks indicating genes and repeats in nos-Gal4>sh-*w* ovaries. **b**, A box plot showing the average log₂-fold ChIP–seq enrichment over input of H3K27me3 (red), H3K9me3 (orange) and Rhi (blue) at the indicated piSL (*n* = 1 kb bins) in control ovaries. The box plots show the median (center line), with interquartile range (box) and whiskers indicating 1.5× the interquartile range at most). **c**, A Venn diagram showing the overlap between Rhi, H3K9me3 and H3K27me3 ChIP–seq signal across genomic 1 kb bins. Signal was considered to be present if the bin overlapped a MACS2 broad peak (*q* < 0.05, broad cutoff <0.1) across two pooled biological replicates. Enrichment was calculated as an OR between Rhi binding in the presence of indicated factor or without. **d**, A heat map displaying predictive performance measured as the AUC across different chromatin marks (top) and target regions (left). The regions are ordered according to their AUC for HP1a + H3K27me3. **e**, A line graph comparing the ability to identify dual-strand piRNA cluster *42AB* based on the strength of individual features (H3K27me3 or HP1a) and combinations (mean signal for H3K27me3 + HP1a). **f**, As in **e** but showing unistrand piRNA cluster *20A*. FPR, false positive rate; TPR, true positive rate.

histone mark enrichment. To compare different marks, we calculated their classification performance as area under the receiver operating characteristic (ROC) curve (AUC) (Extended Data Fig. 3d). We used published HP1a ChIP–seq[29] as a proxy for H3K9me3 as the H3K9me3 antibody appeared to capture some H3K27me3 signal (Supplementary Note 2). We found that H3K27me3 is an accurate predictor of dual-strand piRNA cluster location (AUC 0.867), similar to the predictive power of HP1a binding (AUC 0.842), but that combining both marks improved classification further (AUC 0.927) (Extended Data Fig. 3e). Of note, we observed that H3K27me3 enrichment correlated with E(z) dependence in piRNA production: clusters *42AB* and *38C* showed strong enrichment in H3K27me3 (Fig. 2d,e), whereas *20A* and *80F* whose piRNA production was E(z) independent displayed lower levels of H3K27me3 (Fig. 2d,f). This points toward a role for both H3K27me3 and H3K9me3 in the specification of certain piSL.

## Rhi binding to a subset of piSL depends on E(z)

The dual-strand cluster *42AB* is the major source of Rhi-dependent piRNAs in germ cells (Fig. 3a), and our data indicate that its piRNA

production is E(z) dependent. Prior work implicates Kipf in guiding Rhi to a subset of piSL, including *80F*, which we find is E(z) independent (Fig. 1d–f,h and Extended Data Fig. 2d–f,h), but not *42AB*[22]. We therefore hypothesized that Rhi binding to piSL that are Kipf independent may depend on E(z). To test this, we compared piRNA production from Rhi-dependent loci (*n* = 632, ≥2-fold piRNA reduction after Rhi loss) in Kipf- or E(z)-depleted ovaries. We observed that E(z), but not Kipf, is required for efficient piRNA production from *42AB* and piSL on chr4, whereas *80F* and most heterochromatic regions are E(z) independent but require Kipf (Fig. 3b,c). Other piRNA clusters, such as *38C*, depend weakly on both Kipf and E(z) (Fig. 3b). Together, this suggests that E(z) and Kipf may provide distinct modes of Rhi recruitment to separate piSL.

The production of piRNAs from dual-strand clusters depends on Rhi and we therefore expect piRNA production patterns to closely reflect Rhi binding[6]. To determine whether E(z) directs Rhi binding to selected piSL, we performed Rhi ChIP–seq in E(z)-depleted ovaries and compared this with Rhi ChIP–seq in Kipf-depleted ovaries[22]. We observed that Rhi binding closely mirrored piRNA production

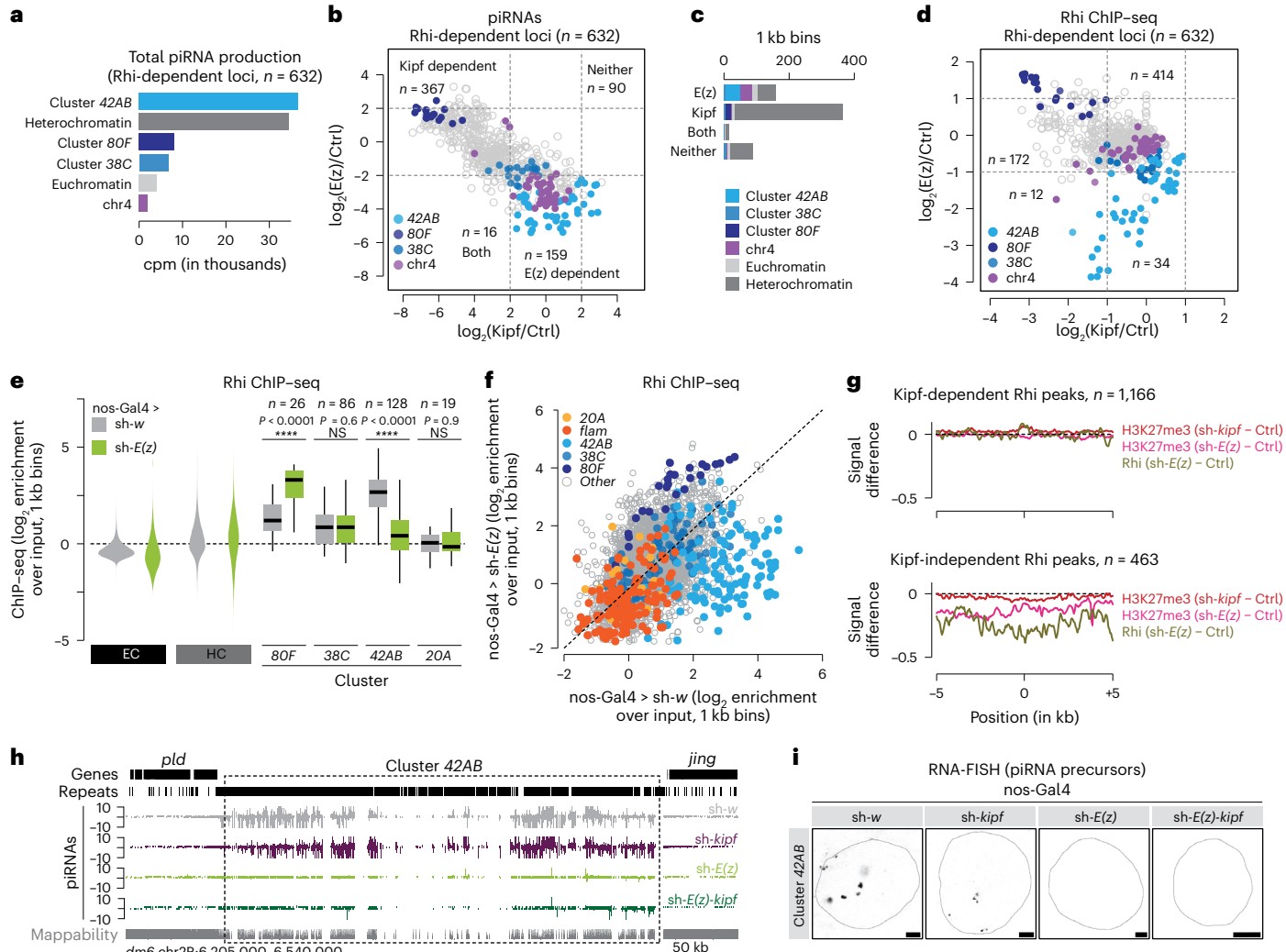

**Fig. 3 | Rhi binding at Kipf-independent loci requires E(z). a**, The genomic origin of Rhi-dependent piRNAs. The analysis was done across 1 kb bins ($n$ = 632) that displayed Rhi dependency (>2-fold mean piRNA loss) across several depletion strategies (pTOsk-Gal4 or MTD-Gal4-driven knockdown, and *rhi* knockout; one to two replicates each). **b**, Scatter plots showing $\log_2$-fold mean changes in uniquely mapping piRNAs in 1 kb bins relative to control (Ctrl) across several depletion strategies (nos-Gal4 or pTOsk-Gal4-driven knockdown of *E(z)*, two replicates each; MTD-Gal4-driven knockdown of *kipf* or *kipf* knockout, one replicate each). Bins were categorized into E(z) dependent and/or Kipf dependent if displaying ≥4-fold mean loss of piRNAs. **c**, Genomic origin for 1 kb bins per E(z)/Kipf-dependency category as defined in **b**. **d**, A scatter plot showing $\log_2$-fold change in Rhi ChIP–seq enrichment following depletion of E(z) using nos-Gal4 or Kipf using MTD-Gal4. The dashed lines indicate a twofold mean loss of Rhi. **e**, Violin plots (left) and box plots (right) showing the average $\log_2$-fold Rhi enrichment by ChIP–seq from ovaries with nos-Gal4-driven *E(z)* knockdown versus control (average of two replicates each) in heterochromatin (HC) and euchromatic

chromosome arms (EC), quantified across 1 kb bins (excluding piRNA clusters). Rhi occupancy at indicated piRNA clusters is shown as box plots ($n$ depicts 1 kb bins analyzed). ****$P$ < 0.0001 based on a two-sided Wilcoxon signed-rank test. Box plots show the median (center line), with interquartile range (box) and whiskers indicating 1.5× the interquartile range at most). **f**, A scatter plot of genomic 1 kb bins contrasting average $\log_2$-fold ChIP–seq enrichment of Rhi in ovaries with nos-Gal4-driven *E(z)* knockdown versus control (average of two replicates). **g**, A metaplot showing the mean difference in H3K27me3 and Rhi ChIP–seq signal upon nos-Gal4 driven *E(z)* or *kipf* knockdown across Rhi peaks categorized as either Kipf dependent or not (Methods). **h**, Genome browser tracks of the dual-strand piRNA cluster *42AB* displaying uniquely mapping piRNAs upon the indicated (double-)knockdowns. **i**, Confocal images showing RNA-FISH signal for transcripts derived from *42AB* in control, *E(z)* and *kipf* knockdown as well as in *E(z)-kipf* dKD, using the nos-Gal4 driver. Scale bar, 5 µm. Data are representative of $n$ ≥ 3 independent experiments.

(Fig. 3d). Consequently, upon E(z) depletion, Rhi enrichment increased at Kipf-dependent piSL and decreased at piSL that depend on E(z) (Extended Data Fig. 4a). We next asked whether depletion of E(z) affects the global distribution of Rhi. In agreement with largely unchanged localization by immunofluorescence (Extended Data Fig. 2a), we found that Rhi was not globally redistributed between euchromatin and heterochromatin (Fig. 3e). Instead, we detected a strong reduction of Rhi at specific regions, including the *ey/Sox102F* region and cluster *42AB* (Fig. 3e and Extended Data Fig. 4b,c), whereas other loci, such as the *80F*, accumulated more Rhi (Fig. 3e). Those observations agreed with

changes in H3K27me3 (Extended Data Fig. 4d). Notably, the observed changes in Rhi occupancy (Fig. 3f) followed piRNA precursor levels, which showed reduced levels for cluster *42AB* and an enrichment at *80F* (Fig. 1d,e). Moreover, the observed differences in Rhi binding were not explained by changes in H3K9me3, which remains essentially unaltered (Extended Data Fig. 4b,c,e).

We next analyzed published ChIP–seq data[22] for genome-wide Rhi and Kipf occupancy. We identified 1,629 high-confidence Rhi binding sites using MACS2 ($q$ < 0.05, >4-fold enrichment). Out of these, 1,166 peaks were lost in *kipf* knockdown and were denoted as Kipf

dependent, while 463 peaks were retained and denoted as Kipf independent (Extended Data Fig. 4f). Supporting separate modes of Rhi recruitment, we observed no reduction in H3K27me3 or Rhi occupancy at Kipf-dependent Rhi peaks following E(z) depletion, in line with these loci lacking H3K27me3 marks and suggesting that Rhi is recruited by Kipf in an H3K27me3-independent manner. In contrast, we observed a broad loss of both H3K27me3 and Rhi occupancy at Kipf-independent Rhi peaks (Fig. 3g), while H3K27me3 was not affected by Kipf depletion (Fig. 3g). This suggests a strong correlation between reduced Rhi binding at Kipf-independent sites and the loss of H3K27me3 following *E(z)* knockdown.

In E(z)-depleted ovaries, our polyA-selected RNA-seq data indicated an up to sixfold reduction in mRNA levels of Moonshiner (Moon), a paralog of TFIIA required for piRNA production from the majority of Rhi-dependent piSL[7] (Supplementary Table 2). We also observed reduced Rhi binding at Kipf-independent peaks in published ChIP–seq from *moon* knockdown[7] (Extended Data Fig. 4g). It is thus possible that Rhi loss upon E(z) depletion may in part be driven by reduced Moon levels. However, we consider secondary effects driven by Moon unlikely since its loss was previously reported to lead to a drastic reduction of piRNAs from cluster *80F* and an increase in *38C*-derived piRNAs[7], opposite to what we observe.

Overall, our results provide evidence that H3K9me3 and H3K27me3 coexist on a subset of piSL and appear to be required for Rhi recruitment at these regions. Thus, two non-redundant mechanisms influence Rhi binding to different piSL, namely E(z)-dependent and Kipf-dependent recruitment.

## Kipf and E(z) independently guide Rhi to its targets

To further investigate the two Rhi recruitment mechanisms, we next determined the effect of simultaneous knockdown of *E(z)* and *kipf*. We sequenced small RNAs from *E(z)-kipf* double-knockdown (dKD) and control ovaries. When compared with knockdown of *w*, TE-derived piRNAs were reduced upon *E(z)* knockdown (Extended Data Fig. 5a). Notably, co-depletion of Kipf and E(z) resulted in more severe effects on piRNA levels (Extended Data Fig. 5b). However, while the combined depletion of E(z) and Kipf has mild effects on piRNA populations relative to *E(z)* knockdown alone (Extended Data Fig. 5c), it leads to pronounced effects when compared with only *kipf* knockdown (Extended Data Fig. 5d).

Next, analyzing piRNA levels at piSL revealed a strong reduction of piRNAs from the cluster *42AB* in *E(z)-kipf* dKD and *E(z)* knockdown compared with *kipf* or control knockdowns (Fig. 3h). To quantify these observations, we compared each (double)knockdown to respective controls across 1 kb bins (Extended Data Fig. 5e). As expected, piRNA production from the unistrand clusters *flam* and *20A* remained unchanged. Upon *E(z)-kipf* dKD we observed a 3.7-fold stronger reduction of piRNAs derived from *42AB* compared with E(z) depletion alone (Extended Data Fig. 5e–g). As observed for TE-derived piRNAs, the combined depletion of E(z) and *kipf* had a mild effect on cluster-derived piRNAs compared with *E(z)* knockdown alone, but a stronger effect compared to *kipf* knockdown alone (Extended Data Fig. 5e,h,i). Unexpectedly, while *kipf* knockdown alone resulted in reduced piRNA levels from *80F*, we observed a recovery of piRNA production to near-normal levels in *E(z)-kipf* dKD ovaries (Extended Data Fig. 5e,g). These results suggest that in the absence of both E(z) and Kipf, Rhi can nevertheless bind to piSL *80F*.

Next, using RNA-FISH, we found that, similar to *E(z)* knockdown, no signals for *42AB* were detected in the *E(z)-kipf* dKD ovaries (Fig. 3i). These results confirmed that transcription of *42AB* piRNA precursors relies on E(z) and is Kipf independent. Notably, Rhi, which typically accumulates in large, continuous structures near the nuclear envelope at satellite regions in *kipf* knockdowns[22], becomes redistributed throughout the nucleus in the *E(z)-kipf* dKD, resembling control and E(z)-depleted ovaries (Extended Data Fig. 5j). Overall, these results

highlight distinct, non-redundant mechanisms of Rhi recruitment at different piSL with some relying on E(z) and others on Kipf, but no evidence of interdependence between both recruitment mechanisms.

## Rhi binds H3K9me3/H3K27me3 in a Kipf-independent context

To investigate whether Rhi binding is driven by H3K9me3/H3K27me3 dual-marked chromatin domains, we used S2 cells that lack an endogenous piRNA pathway[30]. Recent work showed that synthetic dual-chromatin reader domains for specific histone marks can be used to detect chromatin modifications, including combinatorial histone codes, through the use of different binders linked in tandem[31]. Rhi, like other HP1-family proteins, contains a CD that binds H3K9me3-modified histone tails[6,14,15,32]. We generated constructs to recognize H3K9me3 (CD[2xCBX1]), H3K27me3 (CD[2xPc]) or H3K9me3/H3K27me3-decorated domains (a bivalent CD[Pc-CBX1]) (Fig. 4a and Extended Data Fig. 6a), and characterized the binding patterns of dual-CD constructs in S2 cells using CUT&RUN. We confirmed that CD[2xCBX1] bound H3K9me3 (Fig. 4b,c), CD[2xPc] bound H3K27me3 (Fig. 4b,d and Extended Data Fig. 6b) and CD[Pc-CBX1] bound regions with both H3K9me3 and H3K27me3 (Fig. 4b,e).

Next, we tested whether the Rhi CD (CD[2xRhi]) recapitulated the bivalent H3K9me3/H3K27me3 reader CD[Pc-CBX1] (Fig. 4b,e). Strikingly, while CD[2xRhi] binding strongly resembled both that of CD[2xCBX1] and the bivalent CD[Pc-CBX1] genome wide (Fig. 4f and Extended Data Fig. 6c,d), within heterochromatic regions only, CD[2xRhi] binding strongly correlated with H3K27me3 levels (Fig. 4g and Extended Data Fig. 6e). In contrast, tandem Rhi CDs containing F48A mutations in the aromatic cage (CD[2xRhiF48A]) essentially recapitulated the IgG control (Fig. 4b,f,g). Although CD[2xRhi] binding was strong already at loci with only H3K9me3 (odds ratio (OR) 4.2, Fisher's exact test), the strongest CD[2xRhi] binding was observed at loci marked with both H3K9me3 and H3K27me3 (OR 9.3), whereas binding was depleted at loci only marked with H3K27me3 (OR 0.13) (Fig. 4h). Of note, regions marked by both H3K9me3 and H3K27me3 showed no presence of the PRC1 deposited mark H2AK118ub generally associated with the presence of H3K27me3 (ref. 25) (Fig. 4b). These results indicate that Rhi recruitment depends on H3K27me3 but is independent of H2AK118ub (Fig. 4b,d). Indeed, a binning analysis revealed that CD[2xRhi] binding clearly differs from that of CD[2xCBX1] and CD[2xPc] through its ability to bind both clusters *80F* and *42AB*, which are strong CD[2xCBX1] targets, and the CD[2xPc]-targeted *38C* region (Extended Data Fig. 6g,h). Taken together, these results show that, similar to CD[Pc-CBX1], CD[2xRhi] preferentially binds to regions decorated with both H3K9me3 and H3K27me3.

To test whether CD[2xRhi] binding in S2 cells was mediated by Kipf, we repeated the CD experiments upon depletion of Kipf. Despite robust depletion of *kipf* (Extended Data Fig. 6i) and construct expression (Extended Data Fig. 6j), we did not observe differences in the binding patterns of any tested CD construct (Fig. 4i and Extended Data Fig. 6k–n). Importantly, upon *kipf* knockdown, CD[2xRhi] binding was indistinguishable from the control knockdown (Extended Data Fig. 6n), hence suggesting that binding by the Rhi CD in this assay is Kipf independent.

## Kipf-independent Rhi binding depends on E(z)

To directly test the role of H3K27me3 in specifying Rhi CD binding, we performed CUT&RUN in S2 cells for CD[2xCBX1], CD[2xPc], CD[Pc-CBX1] and CD[2xRhi] upon control (*siRen*) or *E(z)* knockdown (*siE(z)*). Principle component analysis showed strong similarities between genome-wide CD[2xPc] binding and H3K27me3 distribution, and between H3K9me3 distribution and CD[2xCBX1] binding (Fig. 4i and Extended Data Fig. 7a). Principle component analysis also confirmed, as we observed in earlier experiments (Fig. 4b and Extended Data Fig. 6c), that binding of CD[2xRhi] in control cells more closely resembles that of CD[Pc-CBX1] over CD[2xCBX1] or CD[2xPc] (Extended Data Fig. 7a). Following *E(z)* knockdown, CD[2xPc] distribution was strongly affected, while CD[2xCBX1] and H3K9me3 remained mostly unchanged (Fig. 4i). Notably, both CD[2xRhi] and CD[Pc-CBX1] distribution

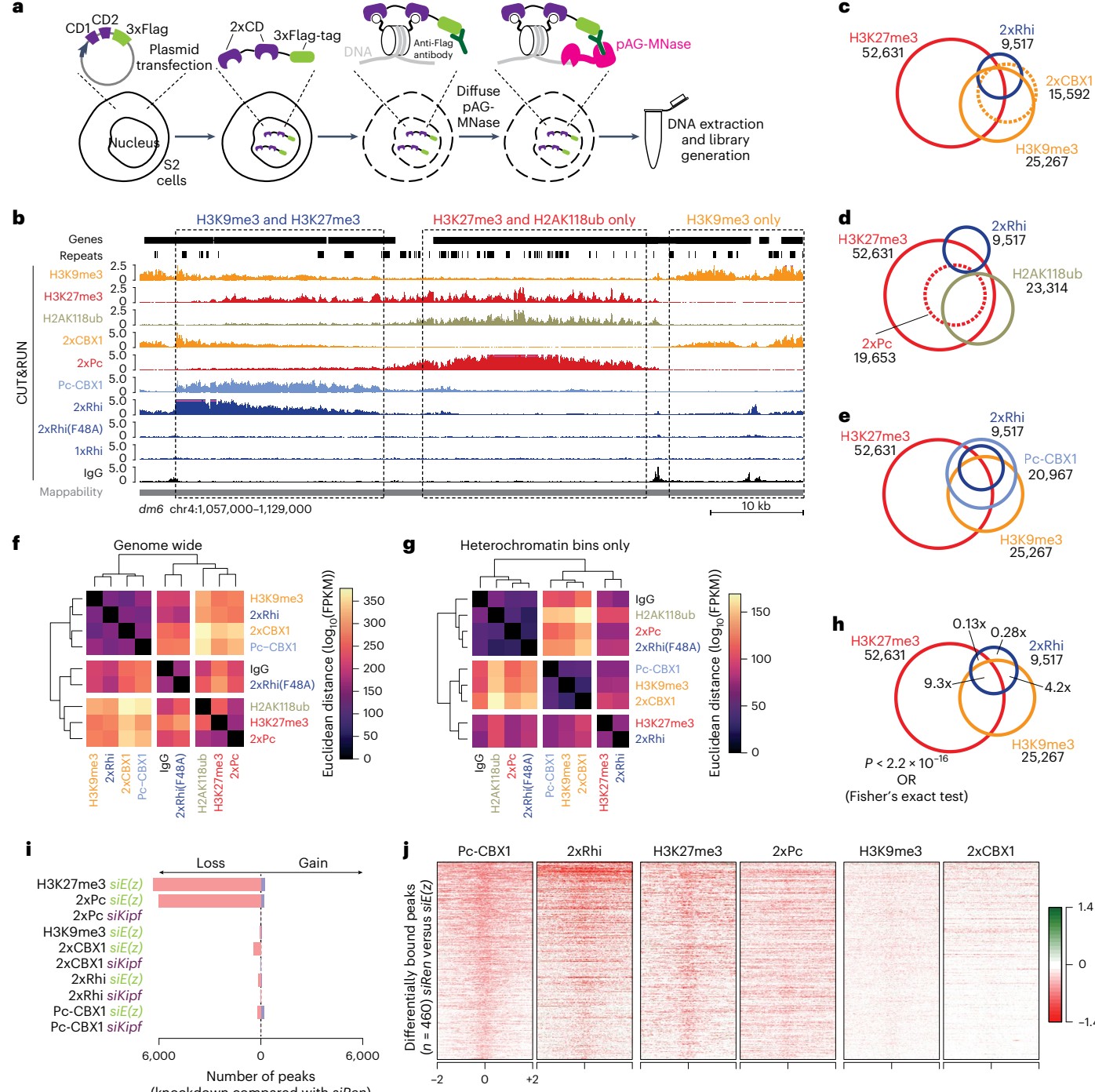

**Fig. 4 | An in vitro system that recapitulates Kipf-independent Rhi binding.**
**a**, A schematic showing the experimental workflow for the CD-binding assay.
**b**, UCSC genome browser tracks of a 72 kb region on chromosome 4 displaying
the binding profiles of the indicated histone modifications and CD constructs
measured by CUT&RUN (cpm across pooled replicates). The dashed boxes
indicate representative areas with either both H3K9me3 and H3K27me3, only
H3K27me3 or only H3K9me3, respectively. The mappability of 100 nt reads is
indicated below the tracks. **c**, A Venn diagram showing overlap between $CD^{2xCBX1}$
binding, H3K9me3, H3K27me3 and $CD^{2xRhi}$ binding across 125,499 genomic 1 kb
bins. Signal was considered to be present if the bin overlapped a MACS2 broad
peak ($q < 0.05$, broad cutoff <0.1) present in at least two biological replicates.
**d**, The same as **c**, but showing $CD^{2xRhi}$ binding, H2AK118ub, H3K27me3 and $CD^{2xPc}$
binding. **e**, The same as **c**, but showing $CD^{Pc-CBX1}$ and $CD^{2xRhi}$ binding, H3K27me3
and H3K9me3. **f**, A heat map and hierarchical clustering (Euclidean distance)

of CUT&RUN signal detected for the indicated CD constructs and histone
modifications ($log_{10}$(FPKM)). The signal is shown as the mean across biological
replicates. Individual replicates shown in Extended Data Fig. 6d. **g**, The same
as **f**, but across a subset of the genome classified as heterochromatin (10,180 out
of 125,499 1 kb bins). Individual replicates shown in Extended Data Fig. 6e. **h**
, The same as **c**, but showing H3K9me3, $CD^{2xRhi}$ binding and H3K27me3. ORs for
$CD^{2xRhi}$ binding to H3K9me3, H3K27me3, both or neither were calculated using
Fisher's exact test. See Extended Data Fig. 6f for all major overlaps. **i**, An overview
of differentially bound regions (called using DiffBind, two-sided Wald test, $P$adj
<0.05) following *kipf* or *E(z)* knockdown. Consensus peaks ($n = 11,053$) were
called using DiffBind with MACS2 broad peaks from the indicated samples.
**j**, Heat maps showing the $log_2$-fold change of *E(z)* knockdown compared to
control knockdown for the indicated constructs and histone modifications
across Pc-CBX1 differentially bound regions.

become more similar to that of CD[2xCBX1] and H3K9me3 in the absence of E(z) (Extended Data Fig. 7a).

To further determine the impact of H3K27me3 on Rhi binding, we divided the genome into 1 kb bins, categorized into six groups based on the degree of H3K27me3 loss upon E(z) depletion. We then plotted for each group the change in binding of the different CDs as assessed by CUT&RUN in *siE(z)* compared with *siRen*. While H3K27me3 loss strongly reduced CD[2xPc] binding, no major genome-wide effects were observed for CD[2xCBX1], CD[Pc-CBX1] or CD[2xRhi] (Extended Data Fig. 7b, top). This reflects the ability of CD[2xPc] to bind H3K27me3 genome wide, whereas only a subset also marked with H3K9me3 can be bound by CD[Pc-CBX1] and CD[2xRhi]. To identify H3K9me3 and H3K27me3 dual-marked regions affected by the E(z) loss, we focused on CD[2xCBX1]-bound regions (reflecting H3K9me3 levels) and observed that E(z) depletion resulted in reduced binding of both CD[2xRhi] and CD[Pc-CBX1]. (Extended Data Fig. 7b, bottom). As expected, we observed no differences when performing the same analysis for *siKipf* (Extended Data Fig. 7c).

We next identified 460 CD[Pc-CBX1] binding sites where binding was significantly reduced (DiffBind, $P$adj <0.05) upon E(z) depletion. Notably, these regions displayed a similar loss of CD[2xRhi] binding, despite retaining comparable H3K9me3 levels (Fig. 4j and Extended Data Fig. 7d). These results indicate that binding of CD[Pc-CBX1] and CD[2xRhi] at these loci require both H3K27me3 and H3K9me3, and that H3K9me3 alone is insufficient. Taken together, these findings suggest that a combinatorial H3K9me3/H3K27me3 histone code, that is independent of Kipf but recognized by the Rhi CD, underlies Rhi binding at a subset of piSL.

## The H3K9me3/H3K27me3 histone code at major piSL is conserved

The *D. melanogaster* and *D. simulans* Rhi CDs have similar binding specificities when expressed in *D. melanogaster* ovaries[33]. This is further supported by the high sequence conservation of the Rhi CD across *Drosophila* species[32]. In particular, aromatic residues in the CD, important for binding to methylated lysine, show high conservation across Drosophilids (Extended Data Fig. 8a). On the basis of this, we hypothesized that Rhi association with H3K9me3/H3K27me3 domains could be conserved.

To test this, we assessed the presence of H3K9me3 and H3K27me3 at piSL in ovaries from *D. simulans* (most recent common ancestor (MRCA) 5.6 million years ago (Ma)), *D. erecta* (MRCA 8.2 Ma), *D. yakuba* (MRCA 11.4 Ma) and *D. ananassae* (MRCA 33.9 Ma) (Fig. 5a). We compared CUT&RUN data for H3K9me3 and H3K27me3 marks with available small RNA-seq of ovaries from these species[4]. We divided the genomes into 10 kb bins that were then classified into 15 groups according to their piRNA levels. We observed that at least for *D. melanogaster* and *D. yakuba*, bins with high levels of piRNAs also showed strong signals of H3K9me3 and H3K27me3, whereas the absence or low levels of piRNAs correlated mostly with the presence of H3K27me3 and absence of H3K9me3 (Fig. 5b,c and Extended Data Fig. 8b). Genome-wide analyses showed that piRNA producing loci are preferentially decorated with both H3K9me3 and H3K27me3 in *D. simulans*, *D. yakuba*, *D. erecta* and *D. ananassae* (Fig. 5d). We also detected both histone marks at the region of the *D. simulans* genome that is syntenic to *42AB* in *D. melanogaster* (Fig. 5e). These results suggest that Rhi binding to H3K9me3/H3K27me3 regions is conserved in Drosophilids.

In conclusion, our study identifies the H3K27 methyltransferase E(z) as an important regulator of TE expression in *Drosophila* germ cells. We show that, in addition to the well-described H3K9me3 mark, H3K27me3 is important for guiding Rhi binding in a Kipf-independent manner, thus contributing to the definition of piSL.

## Discussion

In this study, we provide evidence that co-occurrence of H3K9me3 and H3K27me3, usually thought to occupy distinct domains, plays an important role in specifying Rhi binding to piSL. By performing a screen for factors important for TE control, we identified the H3K27me3 methyltransferase E(z) and demonstrated that several Rhi-bound loci that produce vast amounts of piRNAs in germ cells are decorated with both H3K9me3 and H3K27me3 marks. We demonstrate that at these loci, Rhi recruitment is independent of Kipf and instead requires E(z), probably via its CD. These data demonstrate the existence of two independent modes of Rhi recruitment at distinct subsets of piSL: one that is reliant on the Kipf-recognized DNA sequence and one that is dependent on the chromatin context as established by E(z). Our results also point to the involvement of additional complexes in TE regulation, such as the COMPASS and NSL complex. Of note, a recent study has shown that the germline-specific depletion of subunits of the NSL complex (NSL1, NSL2 and NSL3) resulted in reduced piRNA production from telomeric, thus emphasizing the role of the NSL complex in piRNA precursor transcription within these loci[34].

While previously suggested to be mutually exclusive, our work and that of others demonstrates that H3K9me3 and H3K27me3 coexist in a number of contexts[35–40]. This dual histone code has been reported in the ciliate *Paramecium* where it plays a role in TE regulation[41]. Furthermore, in plants and mammals, H3K27me3 serves as a backup for TE silencing, especially in contexts of low DNA methylation[39,42–46]. Interestingly, regions that give rise to 21U-RNAs in *Caenorhabditis elegans* are located within H3K27me3-rich genomic regions[47] and recruitment of the Upstream Sequence Transcription Complex, required for 21U-RNA precursor transcription, is guided by the CD-containing protein UAD-2, which can bind H3K27me3 (ref. 48). The absence of H3K27me3 decouples UAD-2 from piSL, resembling the effect observed for Rhi upon germline depletion of E(z) in *Drosophila*.

Although E(z)-catalyzed H3K27me3 appears to be required for Rhi recruitment to a subset of piSL, the mechanisms by which E(z) is recruited and when H3K27me3 is deposited at specific *Drosophila* piSL during development remains to be determined. In *Paramecium*, deposition of H3K9me3 and H3K27me3 at TE insertions is mediated by the polycomb protein Ezl1 through interaction with a PIWI protein[49]. Notably, *Drosophila* E(z) has been shown to co-immunoprecipitate with Piwi in ovaries[50], hinting at a possible role of PIWI proteins in guiding E(z) to piSL. Moreover, the presence of H3K27me3 at these loci in both germ cells and in S2 cells that are derived from the somatic lineage suggests that H3K27me3 is deposited maternally[51] or established during early development, and retained as development progresses. A previous study demonstrated divergence in the spatiotemporal expression patterns of Kipf and Rhi, with Kipf exhibiting very low levels in ovarian germline stem cells and cystoblasts[22]. In light of this, we suggest that H3K27me3 may collaborate with H3K9me3 during the early stages of oogenesis, where Rhi functions independently of Kipf. Understanding how and when histone methyltransferases are guided to piSL is critical to better understand the piRNA system and its interplay with chromatin biology.

Our study shows that tandem Rhi CDs have a similar binding pattern as bivalent CD fusion proteins expected to bind H3K9me3/H3K27me3 domains, suggesting that the Rhi CD binds simultaneously to both H3K9me3/H3K27me3. Unlike the HP1a CD, the Rhi CD has been reported to form a homodimer in two crystallographic studies[14,52]. To investigate whether Rhi has the capacity to bind a histone H3 tail carrying dual-modified K9me3/K27me3, we successfully modeled the binding of a Rhi CD dimer to a 40 amino acid histone H3 tail peptide carrying both marks using molecular dynamics (MD) simulation (Extended Data Fig. 8c). However, recent size-exclusion chromatography studies failed to detect Rhi dimerization in solution[23]. Moreover, we were unable to obtain specific binding patterns in S2 cells using a construct carrying a single Rhi CD (CD[1xRhi]) (Extended Data Fig. 8d), possibly arguing against the ability of this construct to dimerize in vivo. However, the binding affinities of individual CD constructs were reported to be insufficient for efficient chromatin profiling[31], hence complicating the interpretation of these results. While our work suggests a role for a dual histone code in determining Rhi binding, probably in

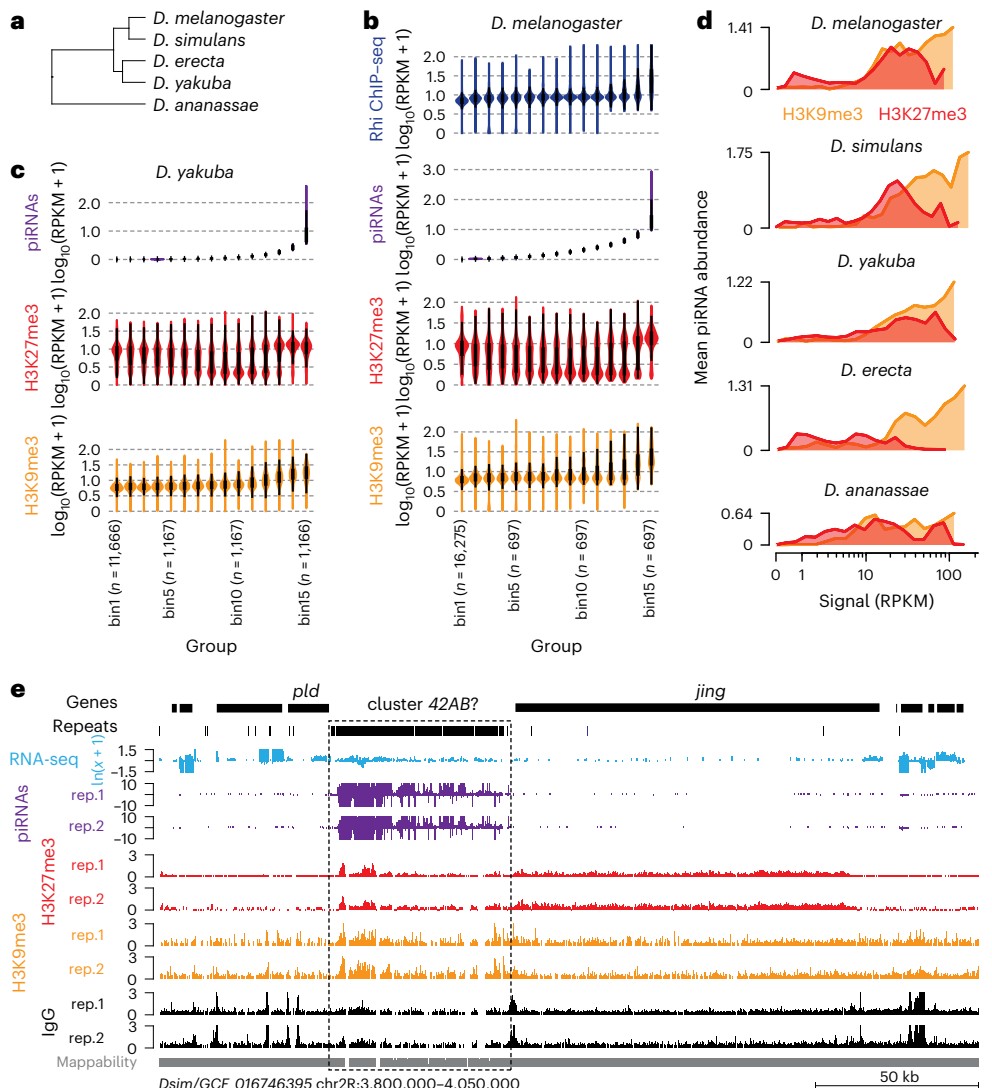

**Fig. 5 | Conservation of Rhi association with H3K9me3 and H3K27me3 in Drosophilids. a**, The phylogenetic tree of the five studied *Drosophila* species. **b**, Violin plots illustrating the distribution of piRNAs (purple), H3K9me3 (orange) and H3K27me3 (red) CUT&RUN signal, and Rhi (blue) ChIP–seq levels (log$_{10}$(RPKM + 1)) in *D. melanogaster* ovaries: 10 kb bins across the genome were divided into 15 groups. Group 1 contains all 10 kb bins with no piRNA expression. The 14 remaining groups contain an equal number of 10 kb bins. Groups are ranked according to piRNA expression levels with the lowest level of the two strands shown to focus on dual-strand regions. The box plots show median (central line), interquartile range (box) and minimum and maximum values (whiskers, 1.5× the interquartile range at most). **c**, The same as **b**, but showing piRNA, H3K9me3 and H3K27me3 levels (log$_{10}$(RPKM + 1)] for *Drosophila yakuba* ovaries. **d**, Line graphs showing mean piRNA abundance (*n* = 3 replicates; signal shown as the lowest of the two strands, log$_{10}$(RPKM + 1)) plotted against H3K9me3 signal (orange, RPKM) and H3K27me3 signal (red) for all 10 kb bins of the indicated *Drosophila* species (*D. melanogaster*, *D. simulans*, *D. yakuba*, *D. erecta* and *D. ananassae*). **e**, UCSC genome browser tracks from *D. simulans* displaying ribo-depleted RNA-seq (light blue), piRNAs (purple), and CUT&RUN for H3K27me3 (red) and H3K9me3 (orange) as cpm at the *42AB*-syntenic region (dashed box). The mappability of 100 nt reads is indicated below the tracks.

a CD-dependent manner, we cannot exclude alternative mechanisms involving an additional co-factor that associates with H3K27me3 or E(z). Considering that previous work found that the Rhi CD does not bind H3K27me3 alone in vitro[6,52], one interesting hypothesis is the presence of a yet-undiscovered Rhi co-factor that links it to H3K27me3. Interestingly, E(z) was recently identified as potential interactor of Deadlock in immunoprecipitation followed by mass spectrometry data from *D. simulans*, *D. erec*ta and *D. virilis*[53], another hypothesis could be that in *D. melanogaster*, this interaction is mediated by the Rhi CD directly. Alternatively, the presence of both H3K9me3 and H3K27me3 could provide a chromatin context (for example, chromatin compaction and/or repression of canonical transcription) more favorable to Rhi binding than H3K9me3 alone.

While bivalent chromatin, characterized by the presence of both H3K4me3 and H3K27me3, or dual domains have been previously suggested to play important roles in regulating gene expression, the diversity of chromatin reader domains able to interpret either each histone post-translational modification or exhibiting dual recognition of both modifications in vitro brings additional complexity to the understanding of chromatin biology[54–61]. In summary, our study provides an unexpected example highlighting in vivo the importance of a combinatorial histone code in the binding specificity of a chromatin-binding protein.

## Online content

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

## Methods

### Fly husbandry and stocks

All flies were kept at 18 °C or 25 °C on standard cornmeal or propionic food. Flies were obtained from the Vienna *Drosophila* Resource Center or from the Bloomington *Drosophila* Stock Center (BDSC). All used fly stocks are listed in Supplementary Table 3. For germline-specific knockdowns, we used a fly line containing a UAS-Dcr2 transgene and a nos-GAL4 driver[62], a pTOsk-GAL4 driver[28] or a nos-GAL4 driver (BDSC:4937), each crossed to stocks expressing short hairpin (sh) RNAs[63] or double-stranded (ds)RNAs[64] under the UAS promoter. After mating at 27 °C for 5 days, parental flies were removed from the vials. Hatching F1 offspring were collected and aged with yeast for 2–3 days before use for follow-up experiments.

### RNA isolation and qRT–PCR

RNA was extracted using TRIzol reagent (Thermo Fisher Scientific) according to the manufacturer's instructions. One microgram of total RNA was treated with DNaseI (Thermo Fisher Scientific), and reverse transcribed using Superscript III (Thermo Fisher Scientific) and oligo(dT)20 primers. Primer sequences are listed in Supplementary Table 4.

### TE screen

One to two different fly lines per gene (one replicate per line) expressing shRNAs or dsRNAs under the UAS promoter[63,64] were crossed with a fly line containing a UAS-Dcr2 transgene and a nos-GAL4 driver, as previously described[62]. RNA was isolated from five to ten ovaries per cross and reverse transcribed as described above. Multiplexed qPCRs were carried out using TaqMan Universal Master Mix II, no UNG (Applied Biosystems) as described[62]. Experiments were performed on a CFX96 Real-Time System C1000 Touch Thermal Cycler (Bio-Rad). $Z$ scores for TE expression were calculated on ΔCT values (CT(TE) – CT(*rp49* control))[65]. The primers and probes used are listed in Supplementary Table 4. Positive hits were confirmed by qPCRs using SYBR green Master mix (Thermo Fisher Scientific). RT–qPCRs were performed on a QuantStudio Real-Time PCR Light Cycler (Thermo Fisher Scientific).

### RNA-FISH

Single-molecule RNA-FISH for transcripts derived from piSL *42AB*, *80F*, *20A* and TEs *Burdock*, *Gypsy12* and *copia* was performed using Stellaris probes (Biosearch Technologies). Probe sequences are listed in Supplementary Table 4. Ovaries from 3–6-day-old flies were dissected in Schneider's *Drosophila* medium and fixed in fixing buffer (4% formaldehyde, 0.3% Triton X-100 and 1× PBS) for 20 min at room temperature (RT), rinsed three times in 0.3% Triton X-100, once in PBS and permeabilized in 70% ethanol at 4 °C overnight. Permeabilized ovaries were rehydrated in wash buffer (10% formamide in 2× SSC) for 10 min. Ovaries were resuspended in 50 µl hybridization buffer (10% dextran sulfate and 10% formamide in 2× SSC) supplemented with 1.5 µl of probes. Hybridization was performed with rotation at 37 °C overnight. Ovaries were then washed twice with wash buffer at 37 °C for 30 min and twice with 2× SSC solution. Then, DNA was stained with 4,6-diamidino-2-phenylindole (DAPI) (1/500 dilution in 2× SSC) at RT for 20 min. Ovaries were mounted in 30 µl Vectashield mounting medium and imaged on a Zeiss LSM-800 confocal microscope. The resulting images were processed using Fiji/ImageJ (v1.54p).

### Immunofluorescence

Fly ovaries were dissected in ice-cold PBS, fixed for 14 min in 4% PFA (Alfa Aesar) at RT and permeabilized with 3× 10 min washes in PBS with 0.3% Triton (PBS-Tr). Samples were blocked in PBS-Tr with 1% BSA for 2 h at RT and incubated overnight at 4 °C with primary antibodies in PBS-Tr and 1% BSA. After 3× 10 min washes at RT in PBS-Tr, secondary antibodies were incubated overnight at 4 °C in PBS-Tr and 1% BSA. After 4× 10 min washes in PBS-Tr at RT with DAPI (Invitrogen) added during the third wash and 2× 5 min washes in PBS, samples were mounted with ProLong Diamond Antifade Mountant (Thermo Fisher Scientific) and imaged on a Leica SP8 or Zeiss LSM-800 confocal microscope. Images were deconvoluted using Huygens Professional or using Fiji/ImageJ (v1.54p). All used antibodies are listed in Supplementary Table 5.

### Molecular cloning and constructs

CD sequences were amplified from cDNA prepared from ovaries or ordered as gBlock fragments from Integrated DNA Technologies and assembled using the NEBuilder HiFi DNA Assembly kit (New England Biolabs E2621) according to the manufacturer's instructions. The final constructs expressed the CDs of interest tagged amino-terminally with an NLS-3xFlag-EGFP cassette under the control of the *D. simulans* ubiquitin promoter. A list and link to the sequence of all constructs used in this study are provided in Supplementary Table 6.

### mRNA-seq (polyA selected)

Total RNA was extracted from 30 ovaries from 3–6-day-old flies using TRIzol (Thermo Fisher Scientific) in three replicates. One microgram of total RNA was subjected to polyA selection and subsequent fragmentation, reverse transcription and library preparation according to the manufacturer's instructions using the Illumina stranded mRNA Prep for sequencing. Sequencing was performed by Novogene on an Illumina Novaseq 6000 instrument.

### Ribo-depleted RNA-seq

Total RNA was extracted from 10–20 ovaries from 3–6-day-old flies using TRizol (Thermo Fisher Scientific) following the manufacturer's instructions. For all the experiments conducted with the pTOsk-Gal4 driver, ribosomal RNA was depleted using RiboPOOL (siTOOLs, Biotech) as described[66]. RNA-seq libraries were produced using NEBNext Ultra Directional Library Prep kit for Illumina, following the manufacturer's instructions for the rRNA depleted RNA. Library size distribution was analyzed on a TapeStation instrument (Agilent Technologies) using a High Sensitivity D1000 ScreenTape. Libraries were pooled in equal molar ratio and quantified with the KAPA Library Quantification kit for Illumina (Kapa Biosystems) and sequenced (paired-end 50) on an Illumina NovaSeq 6000 instrument. For the experiments conducted using the nos-Gal4d driver, rRNA depletion was performed from 1 µg total RNA using the RNA Depletion stranded Library Prep kit (BGI). The samples were then sequenced as paired-end 100 reads on the DNBSEQ G400 sequencer, and adapter-clipped reads were provided by BGI.

### Small RNA-seq

Small RNA extraction was performed as described[4,67]. Argonaute-bound small RNAs were isolated from 10–30 pairs of ovaries from 3–5-day-old flies using TraPR columns (Lexogen, 128.08). Small RNA libraries were generated using the Small RNA-seq Library Prep kit (Lexogen, 052.96) according to the manufacturer's instructions. For all experiments conducted with the pTOsk-Gal4 driver, ovaries of the five *Drosophila* species, and dKD or *kipf* knockdown experiments using the nos-Gal4 driver, sequencing was performed at the CRUK CI Genomics core on an Illumina NovaSeq 6000 instrument. Sequencing for the remaining experiments conducted using the nos-Gal4 driver was performed by Fasteris SA on an Illumina NextSeq550 instrument.

### ChIP–seq

Chromatin immunoprecipitation was performed as previously described[68] with minor modifications. Briefly, 100 ovary pairs were manually dissected into Schneider media and crosslinked in 1% formaldehyde/PBS for 10 min at RT with agitation. The crosslinking reaction was quenched by STOP buffer (PBS 1×, Triton 0.1% and glycine 1 M) and ovaries were washed in PBS and homogenized in a glass douncer: first slightly dounced in PBST 0.1% and centrifuged for 1 min at 400$g$, followed by strong douncing in cell lysis buffer (KCL 85 mM, HEPES

5 mM, NP-40 0.5%, sodium butyrate 10 mM and EDTA-free protease inhibitor cocktail (Sigma)) followed by 5 min centrifugation at 2,000*g*. We performed two washes with cell lysis buffer. The homogenates were then lysed on ice for 30 min in nucleus lysis buffer (HEPES 50 mM, EDTA 10 mM, N lauryl sarkosyl 0.5%, sodium butyrate 10 mM and EDTA-free protease inhibitor cocktail (Sigma)). DNA was sheared using a Bioruptor pico from Diagenode for 10 cycles (30 s on, 30 s off). The sonicated lysates were cleared by centrifugation and then incubated overnight at 4 °C with 5 µl of specific antibodies (Supplementary Table 5). Then 40 µl of Protein A Dynabeads was added and allowed to bind antibody complexes by incubation for 1 h at 4 °C. Following four washing steps with high salt buffer (Tris pH 7.5, 50 mM, NaCl 500 mM, Triton 0.25%, NP-40 0.5%, BSA 0.5% and EDTA pH 7.5, 5 mM), DNA–protein complexes were eluted and de-crosslinked for 10 h at 65 °C. RNA and protein were digested by RNase A and Proteinase K treatments, respectively, before purification using phenol:chloroform:isoamyl alcohol (25:24:1) (Sigma), according to the manufacturer's instructions. Barcoded libraries were prepared using Illumina technology, and subsequently sequenced on a NextSeq High (Illumina) by Novogene (Rhi and H3K9me3 ChIP–seq), or by the Jean Perrin facility (H3K27me3 ChIP–seq).

## CUT&RUN

One million cells per sample were collected and washed three times with wash buffer (20 mM HEPES, pH 7.5, 150 mM NaCl, 0.5 mM spermidine supplemented with protease inhibitors) and resuspended in 1 ml of wash buffer. Then, 10–20 fly ovaries were dissected in ice-cold PBS per sample. Ovaries were digested in 250 µl of dissociation buffer (0.5% Trypsin and 2.5 mg ml$^{-1}$ collagenase A in PBS) for 1 h with shaking at 800 rpm at 30 °C. The digestion was stopped with 250 µl of Schneider medium containing 10% FBS. Cell suspensions were filtered through 40 µm strainers, pelleted and washed three times with wash buffer and resuspended in 1 ml of wash buffer. Following sample preparation, CUT&RUN was performed according to ref. 69 with some modifications. First, 10 µl of activated concanavalin A-coated magnetic beads (Bangs Laboratories) were added to each sample and rotated for 10 min at RT. Bead-bound cells were incubated with 5 µl of antibody (Supplementary Table 5) and 95 µl of antibody buffer (20 mM HEPES pH 7.5, 150 mM NaCl, 0.5 mM spermidine and 0.05% digitonin) overnight at 4 °C. Bead-bound cells were washed twice with Dig-wash buffer (0.05% digitonin in wash buffer). Bead-bound cells were resuspended in Dig-wash buffer containing 1× CUTANA pAG-MNase (Epicypher) and rotated for 1 h at RT. Following pAG-MNase binding, bead-bound cells were washed twice with Dig-wash buffer and resuspended in 100 µl Dig-wash buffer. Chromatin digestion was performed on ice for 30 min by adding 2 µl of 100 mM CaCl$_2$. Digestion was stopped by the addition of 2× STOP buffer (340 mM NaCl, 20 mM EDTA, 4 mM EGTA, 0.05% digitonin, 100 µg ml$^{-1}$ RNase A (Thermo Fisher Scientific) and 50 µg ml$^{-1}$ glycogen) and samples were incubated at 37 °C for 30 min to release DNA fragments into the solution. After centrifugation, 0.1% SDS and 0.2 µg µl$^{-1}$ Proteinase K were added to the supernatant and samples were incubated for 1 h at 50 °C. DNA was extracted using phenol:chloroform:isoamyl alcohol (25:24:1) (Sigma), according to the manufacturer's instructions. Libraries were prepared following the manufacturer's instructions with the NEBNext Ultra II DNA Library Prep kit for Illumina. Sequencing was performed on a NovaSeq 6000 instrument (Illumina).

## Tissue culture, transfection and knockdowns

*Drosophila* Schneider 2 (S2) cells (Thermo Fisher Scientific, R69007) were cultured at 26 °C in Schneider's *Drosophila* media (Gibco) supplemented with 10% heat-inactivated FBS (Sigma). S2 cells were transfected using the TransIT-Insect Transfection Reagent, using 2 million cells per transfection and 1 µg of plasmid DNA, and collected after 48 h. For knockdown experiments, two rounds of electroporation (48 h apart) were performed using the Cell Line Nucleofector kit V

(Amaxa Biosystems; program G-030), as described[70]. CD constructs were cotransfected with 200 pmol of siRNA duplexes (oligo sequences are listed in Supplementary Table 4) during the second nucleofection round or transfected using the TransIT-Insect Transfection Reagent 24 h following the second nucleofection round. Cells were collected 5 days after the first nucleofection.

## Western blot

Cell pellets and ovaries were lysed in RIPA buffer (Pierce) supplemented with protease inhibitors (Roche) and incubated for 30 min at 4 °C. The protein concentration of cleared lysates was quantified using a Direct Detect device (Merck). First, 20 µg total protein was separated on a NuPAGE 4–12% Bis–Tris denaturing gel (Thermo Fisher Scientific) and transferred to a nitrocellulose membrane using an iBLot2 dry transfer (Invitrogen). Primary antibody incubations were performed overnight at 4 °C and secondary antibodies were incubated for 1 h at RT. All antibodies and concentrations used are listed in Supplementary Table 5. Images were acquired using an Odyssey M Imaging system (LI-COR) and processed in Image Studio Lite (LI-COR).

## Processing of RNA-seq

Adapters were removed using Trim Galore! (v0.6.4, --stringency 6) with additional parameters '-a CTGTCTCTTATA --clip_R1 1 --clip_R2 1 --three_prime_clip_R1 1 --three_prime_clip_R2 1' if appropriate. The resulting reads were mapped to *dm6* using STAR (v2.7.3a, --outMultimapperOrder Random --outSAMmultNmax 1 --outFilterMultimapNmax 1000 --winAnchorMultimapNmax 2000 --alignSJDBoverhandMin 1 --sjdbScore 3) and a genome index built using NCBI RefSeq. Gene expression was quantified using feature-Counts (v1.5.3, -s 2 -O --largestOverlap -Q 50) with gene models from Ensembl (release 97) or annotations corresponding to each TE consensus sequences (RepBase, downloaded 20 April 2022).

## Processing of sRNA-seq

The analyses included nos-driven *E(z)*, *kipf*, *w* knockdown (two to three replicates each), nos-driven *E(z)+w* and *E(z)+kipf* double-knockdown (two replicates each), pTOsk-driven *E(z)*, *rhi* or *w* knockdown (two replicates each), and previously published[22]. Gal4-driven *kipf*, *rhi* or *w* knockdown (one replicate each) and *Kipf* knockout, *rhi* knockout and *w1118* flies (one replicate each). Adapters were removed using Trim Galore! (v0.6.4, --stringency 6 --length 18 --max_length 29 -q 0). For previously published small RNA-seq (sRNA-seq), we further used '-a' to specify the adapter, and '--clip_R1' and '--three_prime_clip_R1' to remove non-standard adapters and random nucleotides at the read ends (see Supplementary Table 7 for details). The resulting reads were mapped to *dm6* using Bowtie (v1.2.3, -S -n 2 -M 1 --best --strata --nomaqround --chunkmbs 1024 --no-unal). Gene expression was quantified using featureCounts (v1.5.3, -s1 (or -s 2) -O --largestOverlap -Q 40) with gene models from Ensembl (release 97) or annotations corresponding to each TE consensus sequences (RepBase, downloaded 20 April 2022).

## Differential gene expression and piRNA abundance analysis

RNA-seq differential expression analyses were performed using the DESeq2 (v1.26.0) package from R/Bioconductor. For gene expression, we first applied the DESeq2 function with default parameters followed by fold change shrinkage using the 'ashr' method. To analyze TE expression, we performed a similar analysis but used the size factors previously derived based on gene expression, while dispersion and fold change shrinkage were estimated based on both gene and TE expression. Reads per kilobase per million mapped reads (RPKM) values were calculated following the robust median implementation in DESeq2's 'fpm' function. *P* values adjusted for multiple testing (*P*adj) were obtained using the Benjamini–Hochberg procedure. To consider a gene or TE to be differentially expressed, we required a fourfold change and *P*adj <0.05 unless otherwise specified.

sRNA-seq was analyzed in a similar way, except that the analysis was restricted to reads of length 23–30 nt mapping sense or antisense, respectively, to the TE consensus sequences. Size factors were estimated separately using the estimateSizeFactors function with siRNAs mapping antisense to annotated genes as input.

### Processing of CUT&RUN data

CUT&RUN data were generated in one to four replicates per condition as specified in Supplementary Table 7. Sequencing adapters were removed using Trim Galore! (v0.6.4, --paired --stringency 6). The resulting reads were mapped to the *dm6* reference genome using Bowtie (v1.2.3, -S -y -M1 --best --strata --fr --minins 10 --maxins 600 --chunkmbs 2000 --nomaqround), reporting at most one hit for each read and considering insert sizes between 10 and 600 nt. PCR duplicates were removed using MarkDuplicates from Picard tools (v2.21.2). All the following analyses used the deduplicated data, except for the genome browser visualization.

### Processing of ChIP–seq data

ChIP-seq reads were aligned to the dm6 genome using Bowtie2 (v2.4.2), with the alignment process set to report at most one hit for each read. In case of alignments with the same MAPQ score, the best alignment was randomly selected from among those equally scored alignments. Peaks were called using MACS2 (v2.2.7.1) to capture narrow (-q 0.05 -g dm) and broad peaks (-q 0.05 -g dm --broad --broad-cutoff 0.1). Only uniquely mapped reads were used for the peak calling. As a control we used Input for each condition.

### Processing of publicly available ChIP–seq data

Some publicly available ChIP–seq libraries used for comparison to CUT&RUN were analyzed slightly differently. Specifically, HP1a ChIP–seq data from ovaries were downloaded from GEO (accession GSE140539)[29]. The libraries were paired-end 2× 101 nt and performed in two replicates. Rhi ChIP–seq samples from control, Rhi and Moon knockout ovaries and two ChIP–seq input samples were downloaded from GEO (accession GSE97719)[7]. The Rhi libraries were paired-end 2× 50 nt and performed in one replicate per condition. The input samples were 2× 100 nt and performed in two replicates. Rhi and Kipferl ChIP–seq samples from controls and Rhi and Kipf MTD-Gal4-mediated knockdown ovaries and corresponding input samples were downloaded from GEO (accession GSE202468)[22]. These data had variable read length (50, 74 or 100 nt) and were processed as single-end 50 nt. An overview of all ChIP–seq samples is available in Supplementary Table 7. Sequencing adapters were removed using Trim Galore! (v0.6.4 or v0.6.6, --paired --stringency 6), or alternatively '--hardtrim5 50'. The resulting reads were mapped to the *dm6* reference genome using either Bowtie (v1.2.3, -S -y -M 1 --best --strata --fr --maxins 500 --chunkmbs 2000 --nomaqround), reporting at most one hit for each read. PCR duplicates were removed using MarkDuplicates from Picard tools (v2.21.2). All the following analyses used the deduplicated data, except for the genome browser visualization.

### Peak calling using CUT&RUN or publicly available ChIP–seq data

Peaks were called using MACS2 to capture either narrow (callpeak -f BAMPE -g dm -q 0.01) or broad (callpeak -f BAMPE -g dm -q 0.05 --broad --broad-cutoff 0.1) peaks. Only uniquely mapped reads were used for the peak calling. As controls we used the corresponding IgG libraries for CUT&RUN, or input libraries for ChIP–seq, except for when Rhi ChIP–seq were compared with CUT&RUN and we used the Rhi knockout ChIP–seq as control. For histone modifications with two or more CUT&RUN replicates and CDs with three replicates, we derived a consensus peak set by first merging all peaks and then excluding peaks that were not supported by at least two replicates. For remaining CUT&RUN samples, we used all MACS2 peaks. Area-proportional

Venn diagrams were created in R using the eulerr (v7.0.2) package to provide the best fit. Some low count overlaps might not be displayed due to geometric constraints. All overlaps are listed in the corresponding Source data files.

### Binning analysis in *Drosophila melanogaster*

The binning analysis was done using either 1 kb or 10 kb bins. For 1 kb bins, the genome was divided into 144,916 non-overlapping 1 kb bins. We excluded bins with less than 20% mappability resulting in 125,519 mappable 1 kb bins. For 10 kb bins, the genome was divided into 29,918 bins using a 10 kb sliding window that moves 5 kb at a time. Bins at the chromosome ends with a size less than 5 kb were removed. We excluded bins with 20% mappability, resulting in 25,865 mappable 10 kb bins. Bins derived from mitochondria were further removed, producing a final set of 125,499 bins of size 1 kb and 25,862 of size 10 kb for CUT&RUN. When integrating data from multiple assays, selection of mappable bins was done with respect for the CUT&RUN libraries, and the same bins were used for RNA-seq and sRNA-seq analyses. For binning analysis including only sRNA-seq or RNA-seq, we re-calculated mappability using bowtie alignment with either 100 nt or 26 nt read lengths. CUT&RUN and publicly available ChIP–seq signal was quantified per 50 nt window using the bamCompare module from deepTools (v3.3.2, –binSize 50 --ignoreForNormalization chrM -p 4 --scaleFactorsMethod SES --extendReads --centerReads --exactScaling --minMappingQuality 255 -of bedgraph --operation subtract --pseudocount 0) using pooled IgG CUT&RUN samples as background, except for Rhi ChIP–seq where we used the *rhi* knockout ChIP–seq as background. Next, values below zero were set to zero, before the average normalized signal per window was calculated using bedtools map. RNA-seq and sRNA-seq signal was quantified in a stand-specific manner by converting uniquely mapped reads to BED format and counting the number of reads from each strand falling into a bin with at least half of their length (bedtools intersect, -c -F 0.5).

### Genome browser visualization

For CUT&RUN, RNA-seq and sRNA-seq, we first counted the number of all and uniquely mapped reads, respectively, using samtools, per strand if applicable. Uniquely mapped reads were converted into bigWig files using the deepTools bamCoverage module (v3.3.2, --binSize 1 --ignoreForNormalization chrM --normalizeUsing CPM --exactScaling --scaleFactor $s$ --skipNonCoveredRegions --minMappingQuality 255), where the scale factor, $s$, was set to the number of uniquely mapped reads divided by all mapped reads. Additionally, we used '--extendReads --centerReads' for CUT&RUN samples to center the reads at fragments midpoints, '--filterRNAstrand' for RNA-seq and sRNA-seq to separate the two strands, and '--minFragmentLength 23 --maxFragmentLength 30' to select piRNAs for sRNA-seq. Uniquely mapped reads were converted into bigwig files using deepTools bamCoverage function (v3.5.0 --binSize 1 --ignoreForNormalisation ChrM --normalizeUzing CPM --extendReads --centerReads --skipNonCoveredRegions).

### Assessing classification performance using AUC

Classification performance for individual histone marks or combinations was assessed using an ROC curve. Positive instances were defined as bins overlapping the cluster(s) of interest, with the remaining bins considered negative. The cumulative distribution was calculated in R using the 'cumsum' function with area under the curve calculated using the trapezoidal method.

### Assessing Kipf dependency of Rhi peaks

Rhi ChIP–seq peaks were derived using MACS2 as described above. To focus on the most reliable binding sites, we merged peaks with at least threefold enrichment over background in individual control knockdown replicates into a high-confidence peak set. Any peak that was located to an unplaced contig or that was also present in *rhi* knockdown was excluded from the analysis. Finally, the remaining peaks

were subdivided into Kipf-independent ($n = 463$) or Kipf-dependent ($n = 1{,}166$) peaks, depending on whether they overlapped a Rhi peak in Kipf knockdown or not.

## Coverage plots at Rhi peaks

To visualize read coverage over Rhi peaks (Fig. 3g) we used the deep-Tools computeMatrix module (v3.3.2, reference-point --bin-size 50 -b 5000 -a 5000 --missingDataAsZero --reference-point center). The signal in the peak region was shown as difference between the knockdown and the control conditions.

## Contribution of E(z) and Kipf, respectively, to piRNA production

Analysis was performed per 1 kb bin across the genome. First, we identified 632 bins where piRNA production depends on Rhi (>2-fold reduction in counts per million (cpm), across pTOsk-Gal4- or MTD-Gal4-driven knockdown, and *rhi* knockout; one to two replicates each). Next, we calculated the change in piRNA abundance in E(z)- or Kipf-depleted ovaries (nos-Gal4- or pTOsk-Gal4-driven knockdown of *E(z)*, two replicates each; MTD-Gal4-driven knockdown of *kipf* or *kipf* knockout, one replicate each). Bins with >4-fold reduction in ovaries depleted for Kipf, E(z) or both were considered to be Kipf dependent, E(z) dependent or dependent on both.

## Correlation heat maps

Heat maps were produced using the R package pheatmap (v1.0.12) with Euclidean distance.

## Analysis of CUT&RUN in *E(z)* knockdown samples

To study the genome-wide effect of *E(z)* or *kipf* knockdown in S2 cells, we represented the genome as 117,300 1 kb bins and divided it into six equally sized groups based on the change in H3K27me3 signal upon E(z) loss. Four groups represented variable levels of H3K27me3 loss, one group represented no change and one group displayed a relative gain in H3K27me3 signal. Next, across each group, we calculated the change in 2xPc, 2xCBX1, 2xRhi and Pc-CBX1 binding affinity. Although 2xPc was strongly responsive to H3K27me3 loss, the other chromatin binders were largely unaffected. Hence, we next restricted the analysis to 46,583 bins across the six groups with H3K9me3 (defined as $CD^{2xCBX1}$ above 90th percentile of euchromatic regions).

## Binding affinity analysis using CUT&RUN data

Differential binding affinity analysis was performed using DiffBind (v3.8.4) applied on the MACS2 narrow or broad peaks from *siRen* and *siEz* samples, using the corresponding IgG libraries as controls. For the analysis, we used the dba, dba.count, dba.normalize, dba.contrast and dba.analyze modules with default options. This created a consensus set of peaks present in at least two samples, and then re-quantified the signal intensity at each consensus peak, and re-centered and re-sized the peaks to a 401 nt region around their maxima, resulting in 32,042 narrow or 11,053 broad consensus peaks. The contrasts were specified as *siE(z)* against *siRen* for each target (H3K9me3, H3K27me3, $CD^{2xPc}$, $CD^{2xCBX1}$, $CD^{Pc-CBX1}$ and $CD^{2xRhi}$).

## Euchromatin and heterochromatin coordinates

We used the following coordinates from Fabry and colleagues[71] to define euchromatin (chr2R: 6460000–25286936, chr2L: 1–22160000, chr3L: 1–23030000, chr3R: 4200000–32079331 and chrX: 250000–21500000) and heterochromatin (chr2R: 1–6460000, chr2L: 22160000–23513712, chr3L: 23030000–28110227 and chr3R: 1–4200000) in *D. melanogaster*.

## Reference genomes

We used the *dm6* assembly for *D. melanogaster* downloaded from the UCSC genome browser and the following assemblies downloaded from NCBI: GCF_003285975 for *D. annanassae*, GCF_003286155 for *D. erecta*, GCF_016746395 for *D. simulans* and GCF_016746365 for *D. yakuba*.

## Genome-wide mappability

To estimate genome mappability, we divided the genome into all possible *n*-mers, where *n* is either 26 (for sRNA) or 100 (for CUT&RUN). Those sequences were then mapped back onto the genome and mappability for each position was estimated as the number of reads overlapping a position, divided by *n*.

## Analysis of CUT&RUN in five *Drosophila* species

CUT&RUN was performed in *D. annanassae*, *D. melanogaster*, *D. erecta*, *D. simulans*, and *D. yakuba* for H3K27me3 and H3K9me3 with two replicates per condition, followed by 2×50 nt paired-end sequencing. Sequencing adapters were removed using Trim Galore! (v0.6.4, --paired --stringency 6 -a GATCGGAAGAGCACACGTCTGAACTCCAGTCAC). The resulting reads were mapped to their respective reference genome using Bowtie (v1.2.3, -S -y -M 1 --best --strata --fr --minins 10 --maxins 600 --chunkmbs 2000 --nomaqround), reporting at most one hit for each read and considering insert sizes between 10 and 600 nt. One H3K9me3 replicate for *D. yakuba* displayed low complexity (1.3 million uniquely mapped reads with estimated library size 1.7 million) and was therefore excluded.

## Analysis of sRNA-seq in five *Drosophila* species

Small RNA-seq was performed in *D. annanassae*, *D. melanogaster*, *D. erecta*, *D. simulans* and *D. yakuba* with three replicates per species. The libraries were sequenced as 2× 50 nt paired-end sequencing, but only the first read was used for the analysis. Trim Galore! (v0.6.4) was used to remove an abundant rRNA (--stringency 30 -a TGCTTGGACTA-CATATGGTTGAGGGTTGTA --length 18 -q 0) and sequencing adapters (--stringency 5 -a TGGAATTCTCGG --length 18 --max_length 35 -q 0). Next, Bowtie was used to exclude reads mapping to known *Drosophila* viruses (v.1.2.3, -S -M 1 --best --strata --nomaqround --chunkmbs 1024) using the '--max' and '--un' options to extract unmapped reads. These reads were mapped to their respective reference genome (-S -M 1 --best --strata --nomaqround --chunkmbs 1024), reporting at most one hit for each read.

## Binning analysis in five *Drosophila* species

For the binning analysis, the genome was divided into either 1 kb non-overlapping bins or 10 kb bins with 5 kb overlap. To estimate the mappability within each bin, we divided the genome into all possible 100-mers and mapped those sequences back onto the genome. Mappability was calculated as the number of 100-mers mapping uniquely within each bin divided by the bin size. Bins with size <5 kb (for 10 kb bins) or with <20% mappability were excluded, leaving between 121,081 and 178,773 1 kb or between 24,687 and 38,112 10 kb bins per species. We next quantified the number of uniquely mapped CUT&RUN or sRNA-seq reads per library mapping ≥50% within each bin, considering the forward and reverse strand separately for the sRNA-seq data. To correct for differences in sequencing depth, mappability, and bin size, the resulting bin counts were normalized by their mappability and converted to fragments per kilobase of transcript per million mapped reads (FPKM) values. Bins from mitochondria were excluded from all analyses.

## Scatter plots and violin plots in five *Drosophila* species

The normalized RPKM values were capped at 300 (or 200 for 10 kb bins) and $\log_{10}$-transformed using a pseudocount of 1. To focus on bins with piRNAs derived from both strands, we represented the sRNA-seq data as minimum signal on the forward and reverse strands. For H3K9me3 and H3K27me3, we used the mean of two replicates and for sRNA-seq we used the mean of three replicates. For the violin plots, we grouped the bins based on their piRNA level and displayed the piRNA, H3K27me3

or H3K9me3 level within each group. Briefly, we first constructed one group for bins with no piRNAs. Next, we sorted all remaining bins by piRNA level and extracted 15 equidistant breakpoints, including the lowest and highest piRNA level. Each pair of consecutive non-identical breakpoints were used to construct a piRNA level interval, resulting in between 11 or 14 additional groups per species (1 kb) or between 14 and 15 additional groups (10 kb). All scatter plots and violin plots were constructed in R (v3.6.2) using base graphics, imageScale from sinkr (v0.6), gridExtra and ggplot2.

### Structure modeling of Rhi dimer using AlphaFold multimer

AlphaFold v2.3.2 Colab was used to generate a prediction model[72]. The full-length Rhi CD (amino acids 20–90) reported in the crystal structure of **4U68** was used as input twice to generate the multimer[52]. The structure of the Rhi CD dimer was found to resemble the original structure of **4U68** well[52].

### Structure modeling of histone 3 peptide using Zdock

Zdock was used to evaluate potential binding modes of the peptide to the dimer. To generate the structure of the input histone 3 peptide, the existing crystal structure of the K9me3- and K27me3-containing peptides in **4U68** and **1PFB** were used[52,73]. Within the structure of **4U68**, information for the structure of KQTARK(Me3)S, of the K9me3 section of the histone peptide was available. The structure of LATKAAR(Me3) SAP, of the K27me3 section of the histone peptide was available via **1PFB**. A structure alignment was done using **4U68** and **1PFB** via PyMOL, and the connecting chain of 'TGGKAPRKQ' was added using the Modeller tool in UCSF Chimera 1.16 (refs. [74],[75]). This peptide was then docked using Zdock server, with ZDOCK 3.0.2 (ref. [76]) to yield a predictive model of the binding motif.

### MD simulations

The structural prediction from AlphaFold multimer was used as a basis for MD simulations. MD simulations in explicit water were performed using the GPU accelerated code (PMEMD) of the Amber 16 and Amber-Tools 20 packages[77,78]. Protonation states were calculated using the PDB2PQR server. For the protein scaffold, an evolved version of the Stony Brook modification of the Amber 99 force field (ff14SB)[79] parameters were applied (see below for additional non-standard parameters). TIP3P parameters were assigned to water molecules[80]. To incorporate the non-canonical amino acid K(Me3), into the AlphaFold model, the structure of the K(Me3) residue was used from **4U68** and the partial charges of the K(Me3) residue were set to fit the electrostatic potential generated at the HF/6-31 G(d) level by the restrained electrostatic potential model[81]. The charges were calculated according to the Merz–Singh–Kollman scheme using Gaussian[82]. Each protein complex was immersed in a pre-equilibrated cubic box with a 12 Å buffer of TIP3P water molecules using the Leap module. The systems were neutralized by the addition of explicit counterions ($Na^+$ or $Cl^-$). Long-range electrostatic effects were modeled using the particle mesh Ewald method with periodic boundary conditions[83]. An 8 Å cutoff was applied to Lennard-Jones and electrostatic interactions.

MD simulations were performed according to the following steps: (1) minimization with a maximum cycle of 5,000 and with the steepest descent algorithm for the first 2,500 cycles, with a periodic boundary for constant volume (canonical ensemble, NVT) and without the SHAKE algorithm activated. Positional restraints of 2 kcal mol$^{-1}$ Å$^{-2}$ were applied on heavy atoms of the protein backbone and heavy atoms of the ligand. (2) A 1 ns heating process was performed with a periodic boundary for constant volume (NVT) with SHAKE turned on such that the angle between the hydrogen atoms was kept fixed. Temperature increased from 0 to 300 K in a period of 1 ns with heat bath coupling with a time constant of 2 ps. Positional restraints of 2 kcal mol$^{-1}$ Å$^{-2}$ were applied on heavy atoms of the protein backbone. (3) A 2 ns equilibrium process was performed with a periodic boundary for constant volume (NVT) with SHAKE turned on such that the angle between the hydrogen atoms was kept fixed. An Andersen-like temperature coupling scheme is used to maintain the temperature at 300 K. (4) A 2 ns equilibrium process was performed with a periodic boundary for constant pressure (isothermal–isobaric ensemble, NPT) and with a constant temperature of 300 K, maintained using Langevin dynamics with the collision frequency of 5 ps$^{-1}$. (5) A 100 ns equilibrium process was performed with a periodic boundary for constant pressure (NPT) and with a constant temperature of 300 K. (6) A 1,000 ns production was performed with a periodic boundary for constant pressure (NPT) and with a constant temperature of 300 K. A representative frame was obtained from 1 µS MD simulations using Chimera's cluster analysis tool[84].

### Reporting summary

Further information on research design is available in the Nature Portfolio Reporting Summary linked to this article.

### Data availability

Sequencing data generated in this study has been deposited to the GEO under accession (GSE247156). The following data were retrieved from GEO: RNA-seq, sRNA-seq and ChIP–seq data for Rhi and Kipf (accession GSE202468); HP1a ChIP–seq (GSE140542); Rhi ChIP–seq from control, Rhi, and Moon knockout ovaries (GSE97719); sRNA-seq data for *Drosophila* species (GSE225888). The dm6 genome assembly was downloaded from the UCSC genome browser and the following assemblies were downloaded from RefSeq: GCF_003285975, GCF_003286155, GCF_016746395 and GCF_016746365. Source data are provided with this paper.

### Code availability

All analyses are described in Methods. No custom code was developed as part of this study.

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

## Acknowledgements

We thank members of the Brasset and Hannon groups for fruitful discussions. We thank S. Jensen for her help with preliminary RNA-seq analysis. We are grateful to L. Baumgartner and J. Brennecke for sharing antibodies, unpublished data and for scientific discussion. We thank M. Bao from the Life Science Editors for feedback and comments on the paper. We thank the BDSC and the Vienna *Drosophila* Resource Center for the supply of fly stocks. We thank the Scientific Computing, Genomics, RICS and microscopy core facilities at CRUK Cambridge Institute and the CLIC facility at iGReD for support. G.J.H. is a Royal Society Wolfson Research Professor (RSRP\R\200001). This research was funded in whole, or in part, by Cancer Research UK (grant no. G101107 to G.J.H.; grant nos C9685/ A26398, C9685/A27415 and C9545/A29580 to P.C.) and the Wellcome Trust (grant no. 110161/Z/15/Z to G.J.H.; grant no. 226627/Z/22/Z to G.J.H., B.C.N. and S.B.). Work in the Brasset lab was funded by the Agence Nationale pour la Recherche (grant no. CHApiTRE ANR-20-CE12-0005 to E.B., grant no. BiopiC ANR-21-CE12-0022 to E.B., and grant no. EpiTET 699 ANR-17-CE12-0030-03 to E.B.), the La Fondation ARC pour la recherche sur le cancer (grant no. PJA20171206129 to E.B.). This research was financed also by the French government IDEX-ISITE initiative 16-IDEX-0001 (grant no. CAP 20-25 to E.B.). The funders had no role in study design, data collection and analysis, decision to publish or preparation of the manuscript.

## Author contributions

A.A. and E.K. performed all experiments except those stated below. S.B. performed all computational analyses except ChIP–seq analysis, and initial RNA-seq and small RNA-seq analyses, which were performed by Y.R. and S.M.-M., and the MD simulations/modeling, which was performed by M.J. under the supervision of P.C. E.L.E. performed knockdown experiments using pTOsk-Gal4 and generated RNA-seq libraries, J.v.L. generated small RNA-seq libraries. E.K., A.S. and Y.G. performed CUT&RUN from CD constructs in S2 cells, and Y.G. performed western blot analysis. N.G. generated libraries for RNA-seq and prepared probes for RNA-FISH. The project was conceived by A.A., E.K., B.C.N., E.B. and G.J.H., and supervised by B.C.N., E.B. and G.J.H. The data were analyzed and interpret by A.A., E.K., S.B., B.C.N., E.B. and G.J.H. The paper was written by A.A., E.K., S.B., B.C.N., E.B. and G.J.H. with input from all authors.

## Competing interests

The authors declare no competing interests.

## Additional information

**Extended data** is available for this paper at https://doi.org/10.1038/s41594-025-01584-8.

**Correspondence and requests for materials** should be addressed to Benjamin Czech Nicholson, Emilie Brasset or Gregory J. Hannon.

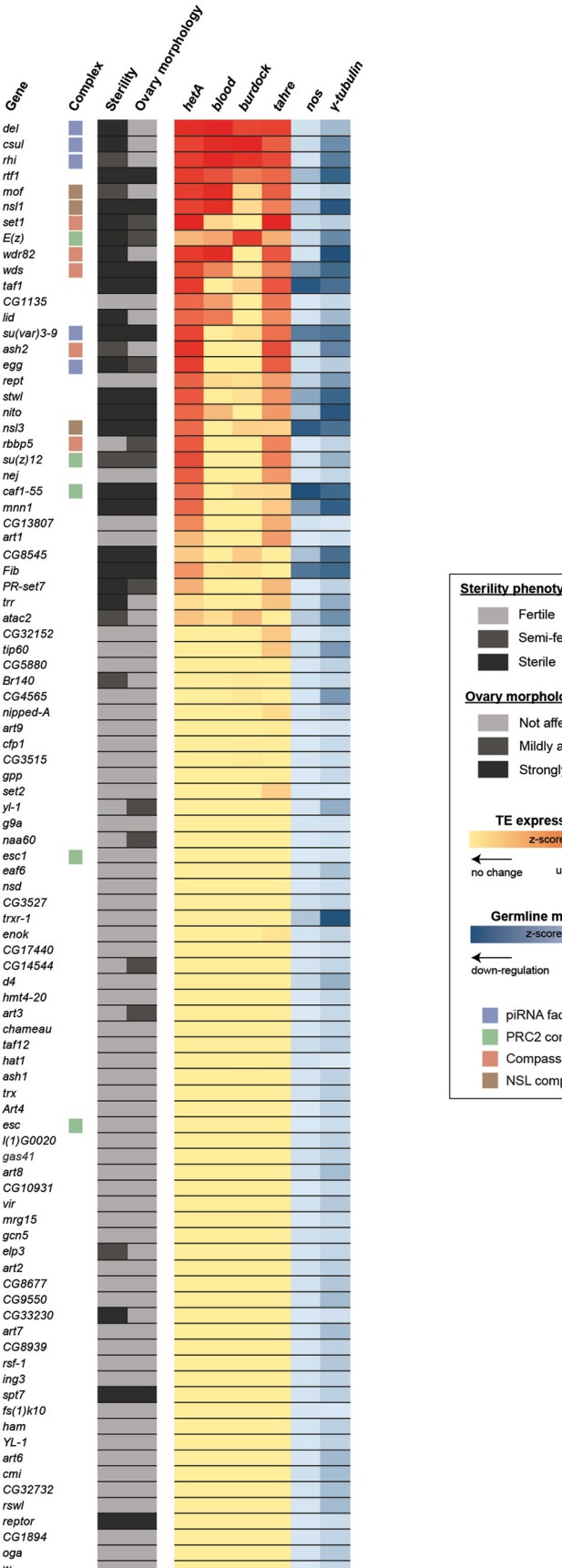

**Extended Data Fig. 1 | Summary of screen results.** Heatmaps summarising gene names, fertility and ovary morphology phenotypes, as well as TE (*hetA, blood, burdock, tahre*) and germline marker (*nos, yTub*) expression, measured by RT-qPCR, upon indicated germline knockdowns (displayed as z-scores). Known piRNA factors and proteins that are part of the COMPASS, PRC2 and NSL complexes are highlighted as indicated in the legend.

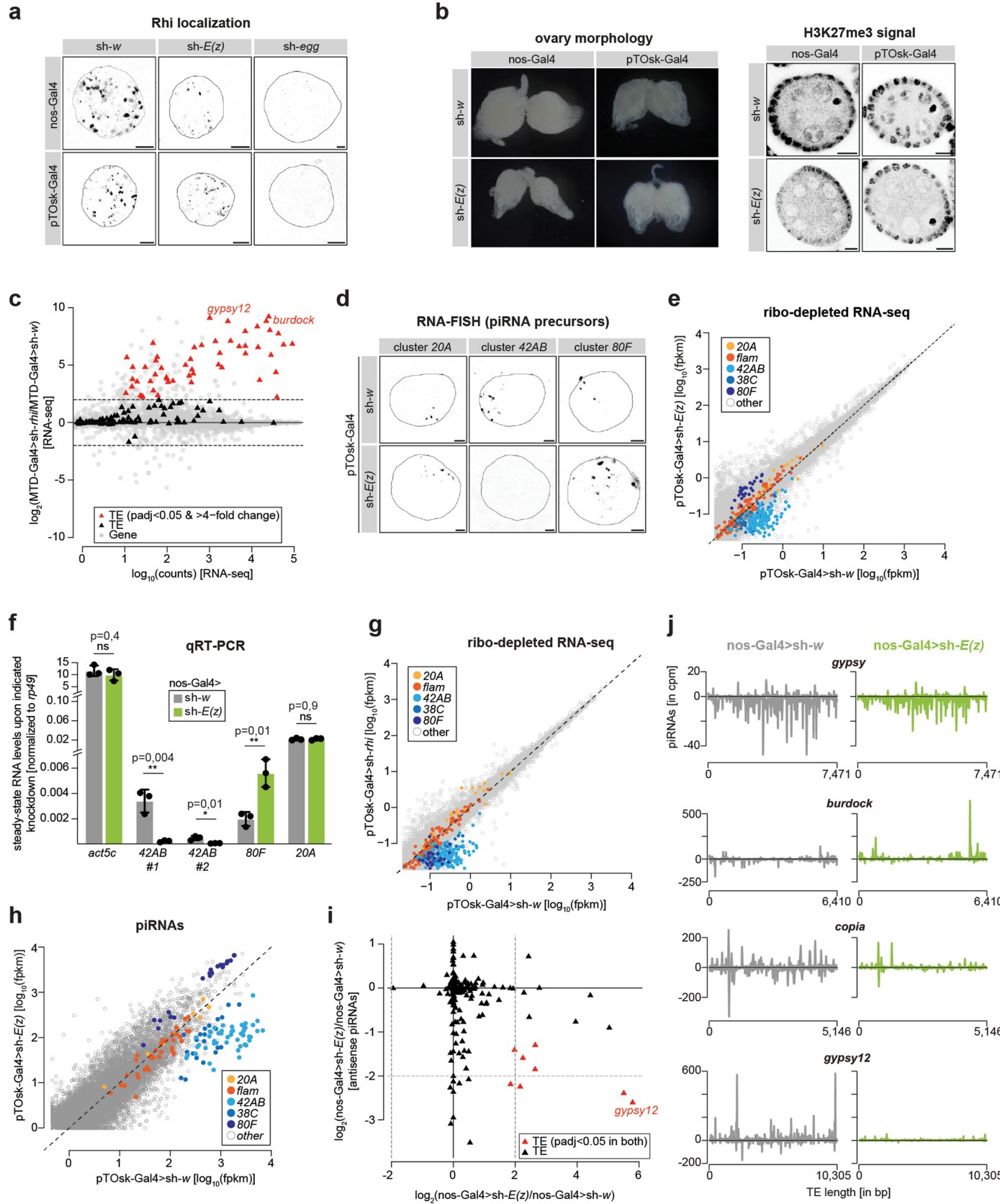

**Extended Data Fig. 2 | See next page for caption.**

**Extended Data Fig. 2 | E(z)-depleted ovaries display piRNA loss and TE derepression. a**, Immunofluorescence staining showing Rhi expression and localisation in nurse cell nuclei of stage 4 egg chambers upon the indicated shRNA and Gal4 driver combination (scale bar: 5 μm). Data are representative of n > =3 independent experiments. **b**, Ovary morphology image (left) and confocal images of egg champers stained with anti-H3K27me3 (right) upon the indicated knockdowns and drivers (scale bars: 10 μm). Data are representative of n > =3 independent experiments. **c**, MA plot showing counts per gene (grey) and TEs (black) in polyA-selected RNA-seq libraries from ovaries of MTD-Gal4 mediated germline knockdown of *rhi* versus control. TEs with padj<0.05 and >4x fold change are shown in red (DESeq2, two-sided Wald test). **d**, Confocal images showing RNA-FISH signal for transcripts derived from piRNA clusters *42AB*, *80F* and *20A* in control and E(z) depleted ovaries using the pTOsk-Gal4 driver (scale bar: 5 μm). Data are representative of n > =3 independent experiments. **e**, Scatter plot depicting normalized ribo-depleted RNA levels (fpkm) of uniquely mapping reads in 1 kb bins in ovaries with pTOsk-Gal4 driven *E(z)* knockdown versus control (average of four replicate experiments each). **f**, Bar graphs showing levels of the indicated piRNA precursor transcripts measure using RT-qPCR upon the indicated germline knockdown. Data are presented as the mean (n = 3) with error bars representing standard deviation. Statistical significance was determined using the unpaired t-test. ** corresponds to P-value < 0.01, * corresponds to P-value < 0.05, and ns corresponds to p > 0.05. **g**, same as e but showing pTOsk-Gal4 driven *rhi* knockdown versus control (average of four replicate experiments each). **h**, Scatter plot depicting normalized piRNA levels of uniquely mapping piRNAs in 1 kb bins in ovaries with pTOsk-Gal4 driven *E(z)* knockdown versus control (two replicates each). **i**, Scatter plot showing TE transcript fold-change against antisense piRNA fold-change in nos-Gal4 driven *E(z)* knockdown versus control. Filled triangles indicate TEs that change on both mRNA and piRNA level (DESeq2, two-sided Wald test, padj<0.05). **j**, Profile plots of piRNA reads over selected TE consensus sequences (*burdock*, *copia*, *gypsy12* and *gypsy*). piRNA counts (normalized to miRNAs) are displayed for indicated genotypes.

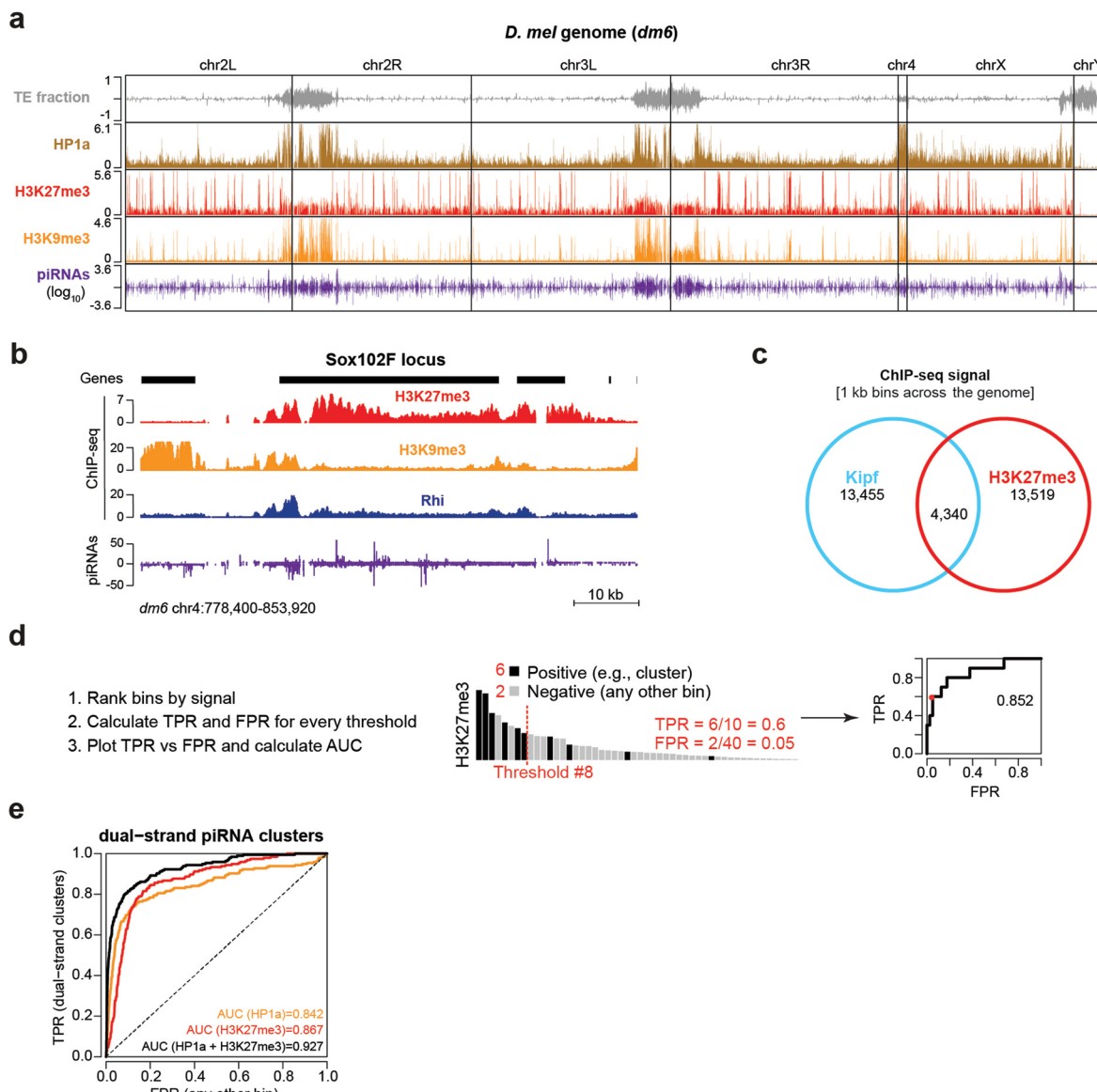

**Extended Data Fig. 3 | Rhi colocalises with H3K9me3 and H3K27me3 in germ cells. a**, Genome-wide view of TE content across *D. melanogaster* chromosomes with TE fraction (grey), HP1a (beige) ChIP-seq (Zenk et al. 2021), H3K27me3 (red) and H3K9me3 (yellow) CUT&RUN and piRNA levels (purple) in wildtype ovaries. CUT&RUN was normalized to IgG signal per 50 nt (see **Methods** for details). **b**, UCSC genome browser tracks of ChIP-seq signal (coverage per million reads) of Rhi (blue), H3K27me3 (red), and H3K9me3 (yellow), and uniquely mapping piRNAs (purple, normalized to miRNA reads) at the *sox102F* locus in nos-Gal4>sh-*w* ovaries. **c**, Venn diagram showing overlap between Kipf and H3K27me3 ChIP-seq signal across genomic 1 kb bins. Signal was considered to be present if the bin overlapped a MACS2 broad peak (q < 0.05, broad-cutoff<0.1) across two pooled biological replicates. **d**, Schematic showing how predictive performance was assessed in a threshold-independent way using an area under ROC curve metric.

First, all genomic 1 kb bins were ranked according to signal strength (for example, highest to lowest H3K27me3). Second, for every possible rank threshold, true positive rate (TPR) is calculated as number of bins of interest (for example, located within a dual-strand piRNA cluster) above the threshold divided by the total number of bins of interest. Similarly, false positive rate (FPR) is calculated as the number of other bins (for example, located outside of a dual-strand piRNA cluster) above the threshold, divided by the total number of other bins. Third, TPR is plotted against FPR and the area under curve (AUC) is calculated. A higher metric indicates more predictive power and AUC at 0.5 indicate performance expected by random guessing. **e**, Line graphs comparing the ability to identify dual-strand piRNA clusters based on the strength of individual features (H3K27me3 and HP1a) or combinations.

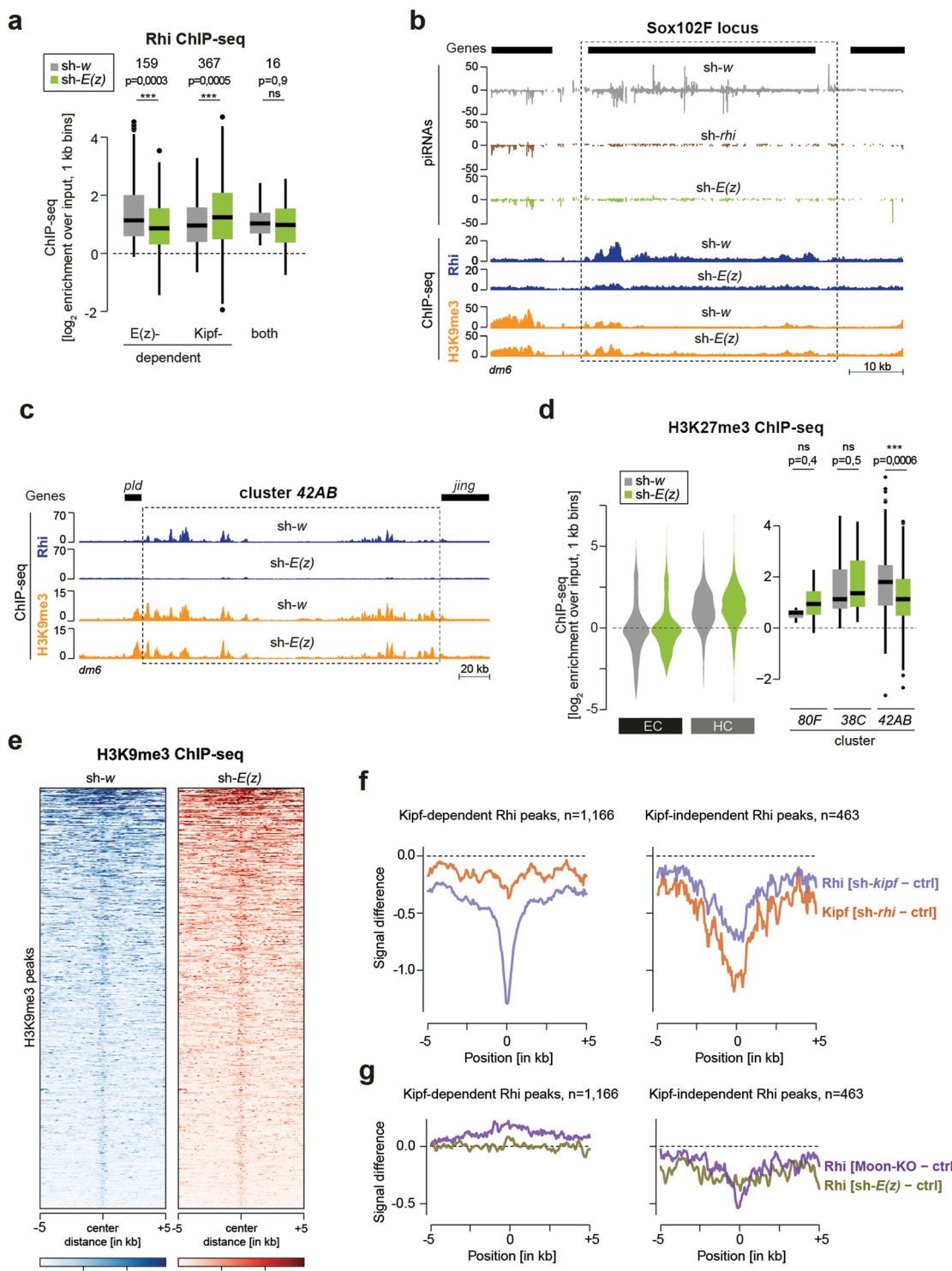

**Extended Data Fig. 4 | See next page for caption.**

**Extended Data Fig. 4 | Rhi binding at Kipf-independent loci depends on E(z). a**, Box plots showing Rhi ChIP-seq enrichment per E(z)/Kipf-dependency category. *** corresponds to P < 0.001, ns corresponds to p > 0.05 based on Wilcoxon signed-rank test. **b**, UCSC genome browser tracks of the *sox102F* locus displaying profiles of small RNAs uniquely mapping in control, E(z)-depleted and Rhi-depleted ovaries and ChIP-seq signal for Rhi and H3K9me3 in control and E(z)-depleted ovaries. **c**, UCSC genome browser tracks of ChIP-seq signal for Rhi and H3K9me3 in control and E(z)-depleted ovaries of the *42AB* dual-strand piRNA cluster. Dashed line indicates approximate piRNA cluster boundaries. **d**, Violin plots (left) and box plots (right) showing average $\log_2$-fold H3K27me3 enrichment by ChIP-seq from ovaries with nos-Gal4 driven *E(z)* knockdown versus control (average of two replicate experiments each) in heterochromatin (HC) and euchromatic chromosome arms (EC), quantified across 1 kb bins (excluding piRNA clusters). H3K27me3 occupancy at indicated piRNA clusters is shown as box plot quantification (n depicts 1 kb bins analyzed for each piRNA cluster). *** corresponds to P < 0.001, ns corresponds to p > 0.05 based on Wilcoxon signed-rank test. Box plots show median (centre line), with interquartile range (box) and whiskers indicating at most 1.5x interquartile range. **e**, Heatmap of H3K9me3 ChIP-seq signal across wildtype H3K9me3 peaks (n = 4,785) in control and *E(z)* knockdown ovaries. **f**, Metaplot showing mean difference in ChIP-seq signal for Kipf (orange) in *rhi* knockdown, and Rhi (light blue) in *kipf* knockdown across Rhi peaks categorised as either Kipf-dependent or not (see **Methods**). **g**, Metaplot showing mean difference in ChIP-seq signal for Rhi in *moon* (purple) knockout or *E(z)* knockdown (yellowish green) across Rhi peaks categorised as either Kipf-dependent or not (see **Methods**).

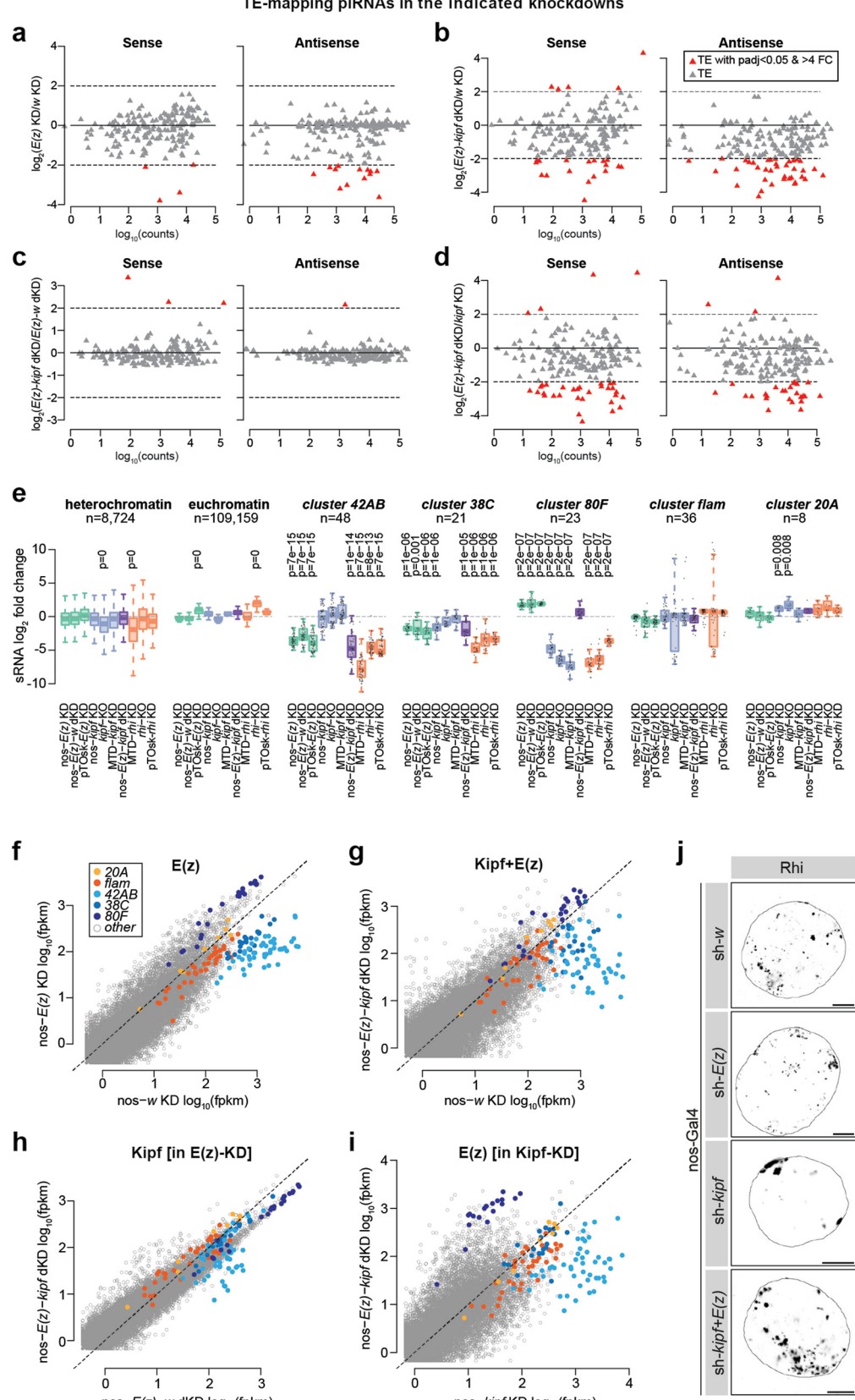

**Extended Data Fig. 5 | See next page for caption.**

**Extended Data Fig. 5 | Rhi binding upon *E(z)-kipf* double knockdown. a-d**, MA plot showing counts per TE (black) in sRNA-seq libraries from ovaries of germline (**a**) *E(z)* KD versus control, (**b**) *E(z)-kipf* dKD versus control, (**c**) *E(z)-kipf* dKD versus *E(z)* KD, and (**d**) *E(z)-kipf* dKD versus *kipf* KD. Sense and antisense piRNAs are shown separately and TEs with padj<0.05 and >4x fold change are shown in red (DESeq2, two-sided Wald test). **e**, Boxplots showing changes in piRNAs mapping uniquely to genomic 1 kb bins within heterochromatin, euchromatin or indicated piRNA clusters (*n* refers to the number of 1 kb bins for each category). Signal shown as a $\log_2$-ratio between the indicated KD and its respective control (average of 1-3 biological replicates). P-values were calculated using a two-sided paired Wilcoxon signed-rank test for >2-fold changes. Boxplots show median (central line), interquartile range (IQR, box), and minimum and maximum values (whiskers, at most 1.5*IQR). **f**, Scatter plot depicting normalized uniquely mapping piRNA levels in 1 kb bins in ovaries with nos-Gal4 driven *E(z)* knockdowns versus control (average of two replicates each). **g**, Scatter plot depicting normalized uniquely mapping piRNA levels in 1 kb bins in ovaries with nos-Gal4 *E(z)-Kipf* dKD compared to nos-Gal4 driven *w* knockdown (average of two replicates each). **h**, Same as d but comparing *E(z)-kipf* dKD to nos-Gal4 driven *E(z)-w* dKD (average of two replicates each). **i**, Same as d but comparing *E(z)-kipf* dKD to nos-Gal4 driven *kipf* knockdown (average of two or three replicates each). **j**, Immunofluorescence staining showing Rhi expression and localisation in nurse cell nuclei upon the indicated shRNA and nos-Gal4 driver combination (scale bar: 5 µm). Data are representative of n > =3 independent experiments.

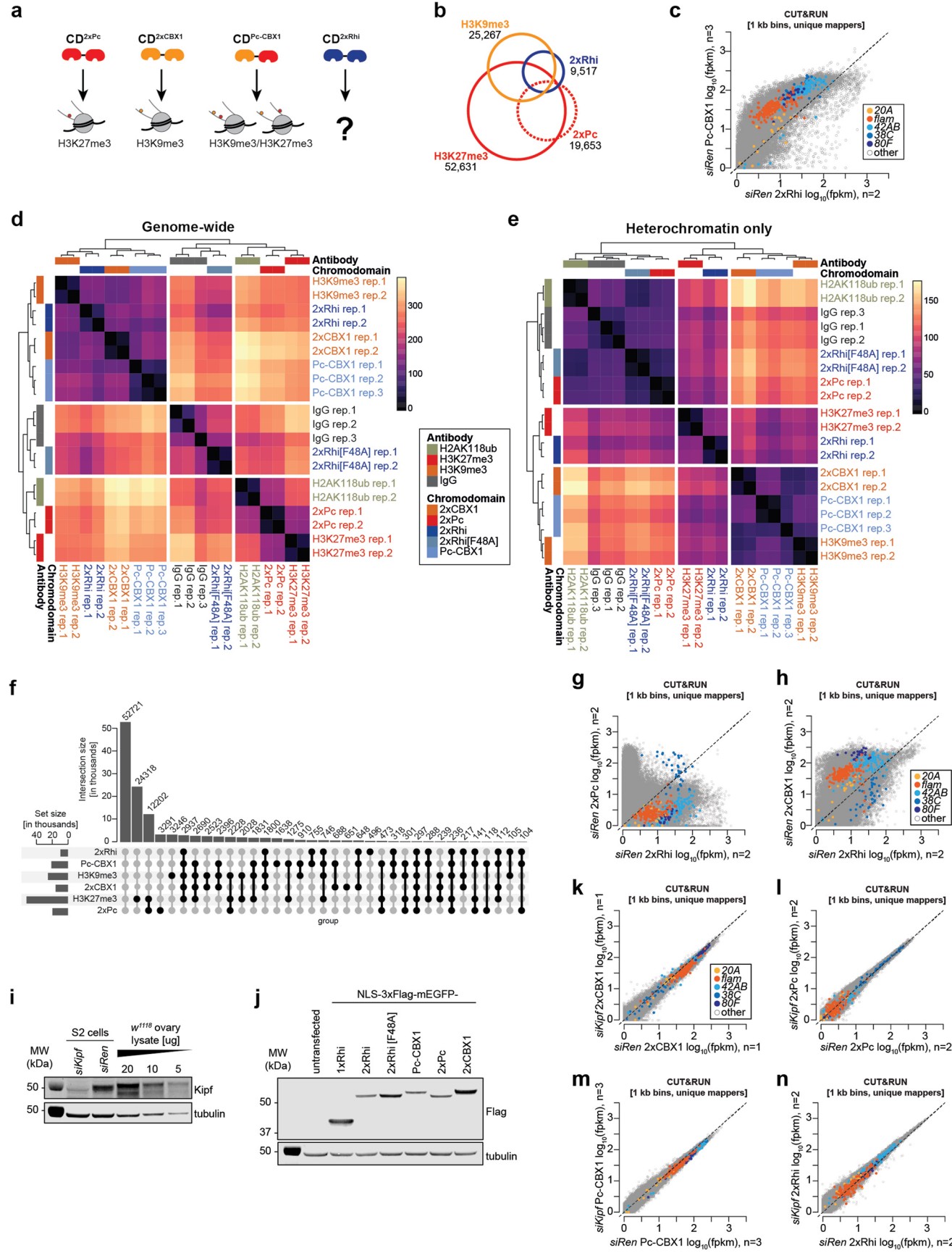

**Extended Data Fig. 6 | See next page for caption.**

**Extended Data Fig. 6 | Rhi chromodomains associate with regions co-occupied by H3K9me3 and H3K27me3. a**, Schematic showing the experimental workflow and cartoon of the dual-CD constructs used in the assay. **b**, Venn diagram showing overlap between $CD^{2xPc}$ binding, H3K27me3, H3K9me3, and Rhi across 125,499 genomic 1 kb bins. Signal was considered to be present if the bin overlapped a MACS2 broad peak (q < 0.05, broad-cutoff<0.1) present in at least two biological replicates. **c**, Scatter plot depicting CUT&RUN signal (fpkm) of uniquely mapping reads in 1 kb bins of $CD^{2xRhi}$ versus $CD^{Pc-CBX1}$ in S2 cells (average of 2-3 replicate experiments each). **d**, Heatmap and hierarchical clustering (Euclidean distance) of CUT&RUN signal detected for the indicated chromodomain constructs and histone modifications [$\log_{10}$(fpkm)]. **e**, Same as d, but across a subset of the genome classified as heterochromatin (10,180 out of 125,499 1 kb bins). **f**, UpSet plot showing all major intersections ( >100 1 kb bins) across the six indicated

CUT&RUN experiments. **g**, as in c but showing $CD^{2xRhi}$ versus $CD^{2xPc}$ (average of 2 replicate experiments each). **h**, as in c but showing $CD^{2xRhi}$ versus $CD^{2xCBX1}$ (average of 2 replicate experiments each). **i**, Western blot analyses showing Kipf expression in S2 cells (treated as indicated) or $w^{1118}$ ovaries. Tubulin is shown as loading control (experiment was repeated twice with similar results). **j**, Western blot analyses showing the levels of the indicated chromodomain constructs (Flag) in S2 cells that were transfected with the corresponding plasmids. Tubulin is shown as loading control (similar results were obtained in two independent experiments). **k**, Scatter plot depicting CUT&RUN signal (fpkm) of uniquely mapping reads in 1 kb bins of $CD^{2xCBX1}$ upon control or *kipf* knockdown in S2 cells (1 replicate experiment). **l-n**, as in k but depicting $CD^{2xPc}$, $CD^{Pc-CBX1}$, and $CD^{2xRhi}$, respectively (2-3 replicate experiments each, as indicated).

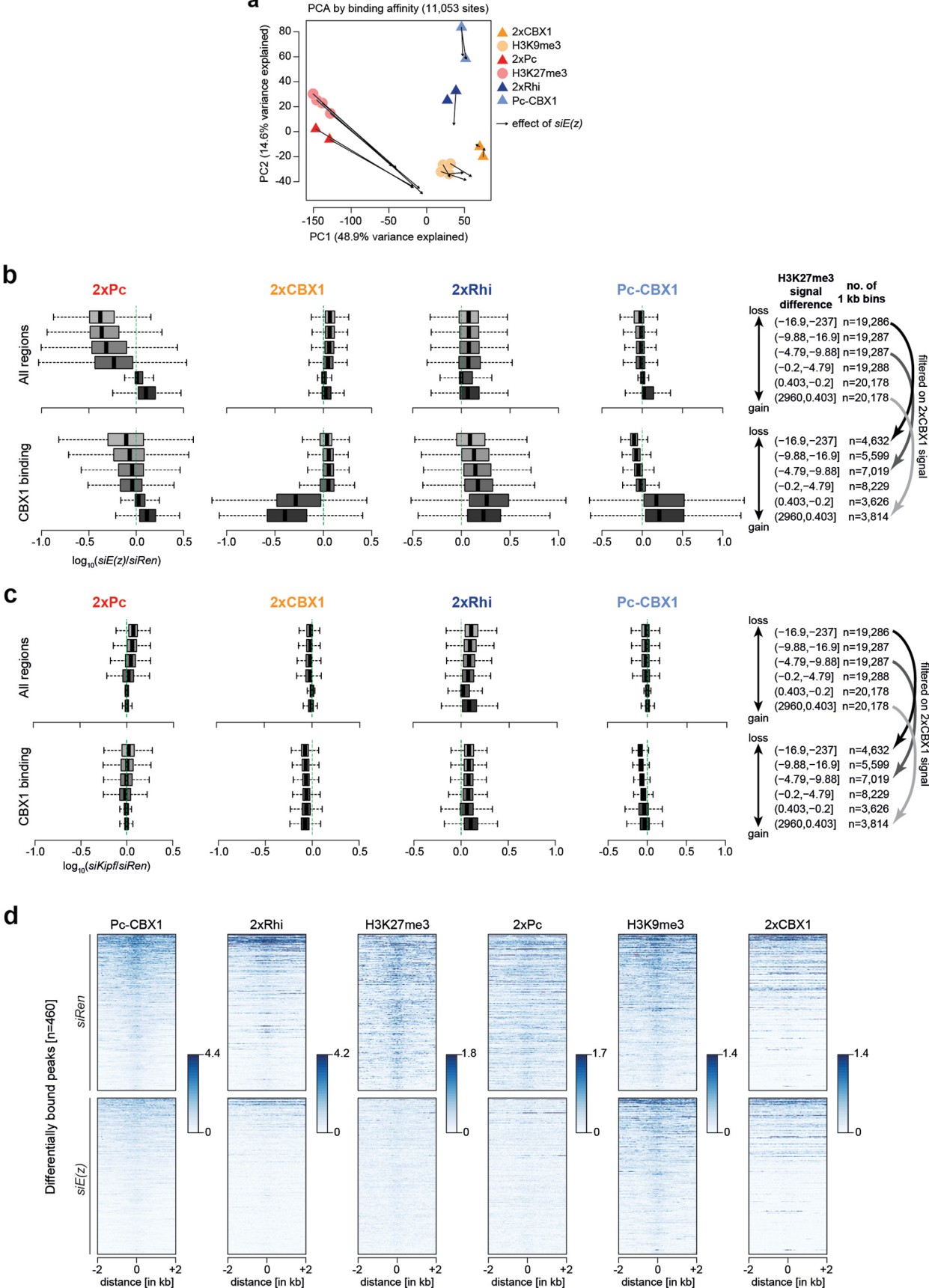

**Extended Data Fig. 7 | See next page for caption.**

**Extended Data Fig. 7 | CD$^{2xRhi}$ binding in S2 cells depends on E(z). a**, Principal component analysis (PCA) plot of the binding affinities of the indicated chromodomain fusion proteins expressed in S2 cells in control condition (circle/triangle) and *E(z)* knockdown (arrowhead). One CD$^{2xRhi}$ sample without arrowhead lacks a corresponding knockdown sample. **b**, Box plot showing loss of binding of the indicated chromodomain fusion proteins in S2 cell upon *E(z)* knockdown compared to control knockdown (*siRen*). Shown are log$_{10}$ fold change per 1 kb bin either genome wide (top, n = 117,300 bins with a non-zero change) or at CD$^{2xCBX1}$-bound regions (bottom, n = 46,583 bins, CD$^{2xCBX1}$ above 90$^{th}$ percentile

of euchromatic regions). The genome-wide bins were divided into six groups according to the severity of H3K27me3 loss (higher to lower, thresholds indicated to the right). Boxplots show median (central line), interquartile range (IQR, box), and minimum and maximum values (whiskers, at most 1.5*IQR). **c**, Same as a, but for *kipf* knockdown (*siKipf*) compared to control knockdown (*siRen*). **d**, Heatmaps showing the indicated chromodomain fusion protein and histone modification CUT&RUN signal in the 4 kb surrounding CD$^{Pc-CBX1}$ differentially bound peaks in control (*siRen*) and *E(z)* knockdowns.

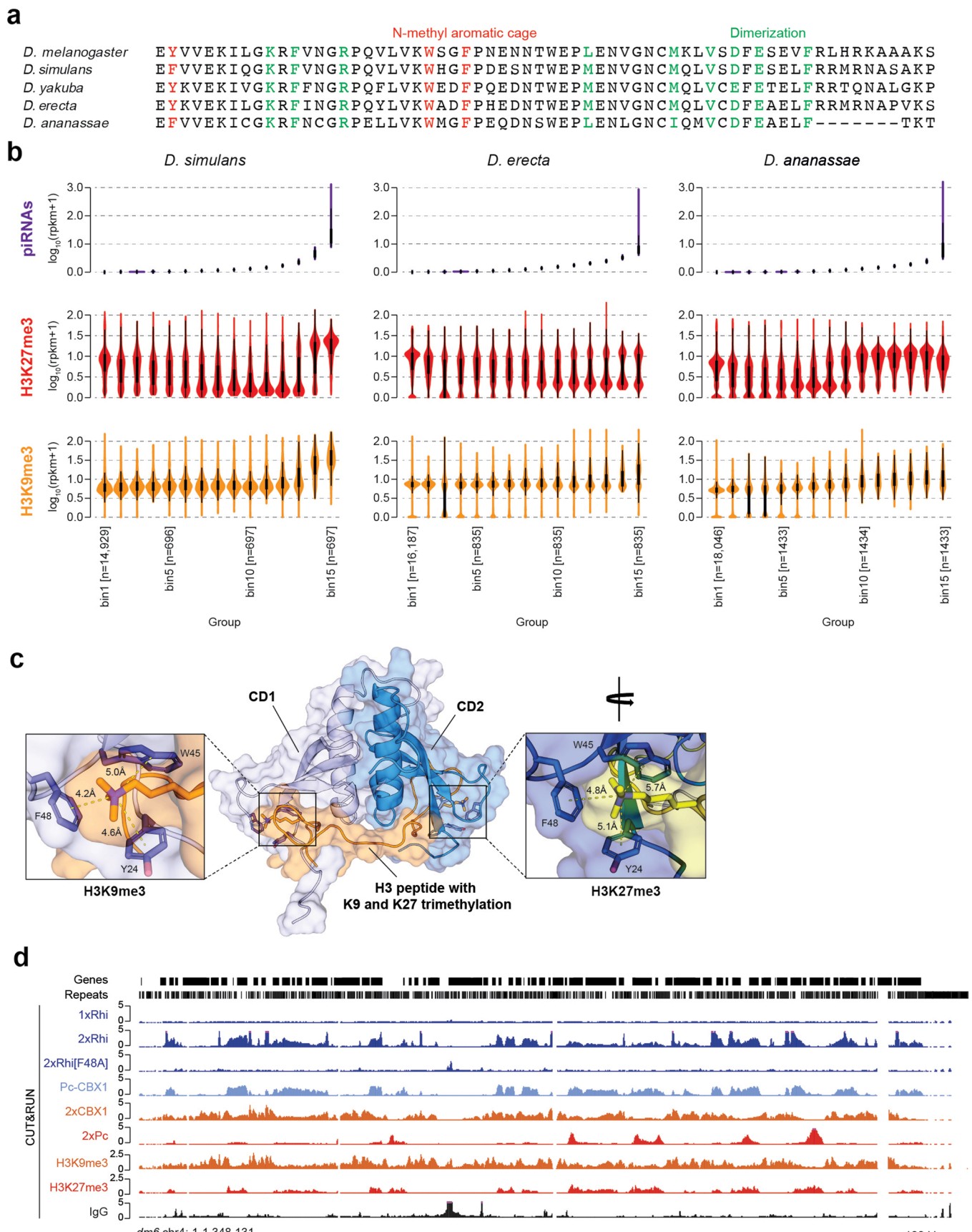

**a**

N-methyl aromatic cage          Dimerization

*D. melanogaster* EYVVEKILGKRFVNGRPQVLVKWSGFPNENNTWEPLENVGNCMKLVSDFESEVFRLHRKAAAKS
*D. simulans* FVVVEKIQGKRFVNGRPQVLVKWHGFPDESNTWEPMENVGNCMQLVSDFESELFRRMRNASAKP
*D. yakuba* YKVEKIVGKRFFNGRPQFLVKWEDFPQEDNTWEPMENVGNCMQLVCEFETELFRRTQNALGKP
*D. erecta* YKVEKILGKRFINGRPQYLVKWADFPHEDNTWEPMENVGNCMQLVCDFEAELFRRMRNAPVKS
*D. ananassae* FVVVEKICGKRFNCGRPELLVKWMGFPEQDNSWEPLENLGNCIQMVCDFEAELF------TKT

**b**

**c**

**d**

Extended Data Fig. 8 | See next page for caption.

**Extended Data Fig. 8 | Dual-strand piRNA producing loci are associated with both H3K9me3 and H3K27me3 in Drosophilids. a**, Protein sequence alignment of the chromodomains of Rhi orthologs across the indicated species. Residues forming the hydrophobic pocket for trimethyl lysine binding are indicated in red. Residues important for chromodomain dimerization are indicated in green. **b**, Violin plots illustrating the distribution of piRNAs, H3K9me3 and H3K27me3 CUT&RUN signal [$\log_{10}$(rpkm+1)] in *D. simulans*, *D. erecta*, and *D. ananassae* ovaries. All 10 kb bins across the genome were divided into 15 groups. Group 1 contains all 10 kb bins with no piRNA expression. The 14 remaining groups contain an equal number of 10 kb bins. Groups are ranked according to piRNA expression levels with the lowest level of the two strands shown to focus on

dual-strand regions. Boxplots show median (central line), interquartile range (IQR, box), and minimum and maximum values (whiskers, at most 1.5*IQR). **c**, Model of the Rhi chromodomain dimer in complex with a histone 3 peptide with trimethylated K9 and K27 residues. Representative frame obtained from 1 μS molecular dynamic simulations show the potential for the dimer's interaction with both H3K9me3 and H3K27me3 to be simultaneous. Zoom-in indicated aromatic residues that interact with the methylated lysine group. **d**, UCSC genome browser tracks of *D. melanogaster* chromosome 4 displaying the binding profiles of the indicated chromodomain constructs measured by CUT&RUN (counts per million [cpm] across pooled replicates). Gene tracks are shown above.

# Reporting Summary

## Statistics

For all statistical analyses, confirm that the following items are present in the figure legend, table legend, main text, or Methods section.

| n/a | Confirmed | |
|---|---|---|
| ☐ | ☒ | The exact sample size (*n*) for each experimental group/condition, given as a discrete number and unit of measurement |
| ☐ | ☒ | A statement on whether measurements were taken from distinct samples or whether the same sample was measured repeatedly |
| ☐ | ☒ | The statistical test(s) used AND whether they are one- or two-sided<br>*Only common tests should be described solely by name; describe more complex techniques in the Methods section.* |
| ☒ | ☐ | A description of all covariates tested |
| ☐ | ☒ | A description of any assumptions or corrections, such as tests of normality and adjustment for multiple comparisons |
| ☐ | ☒ | A full description of the statistical parameters including central tendency (e.g. means) or other basic estimates (e.g. regression coefficient) AND variation (e.g. standard deviation) or associated estimates of uncertainty (e.g. confidence intervals) |
| ☐ | ☒ | For null hypothesis testing, the test statistic (e.g. *F*, *t*, *r*) with confidence intervals, effect sizes, degrees of freedom and *P* value noted<br>*Give P values as exact values whenever suitable.* |
| ☒ | ☐ | For Bayesian analysis, information on the choice of priors and Markov chain Monte Carlo settings |
| ☒ | ☐ | For hierarchical and complex designs, identification of the appropriate level for tests and full reporting of outcomes |
| ☒ | ☐ | Estimates of effect sizes (e.g. Cohen's *d*, Pearson's *r*), indicating how they were calculated |

*Our web collection on statistics for biologists contains articles on many of the points above.*

## Software and code

Policy information about availability of computer code

| Data collection | No software was used for data collection. |
|---|---|
| Data analysis | HTS data processing: FastQC v0.11.8, Trim Galore! v0.6.4 or v0.6.6, STAR v2.7.3a, bowtie v1.2.3, bowtie2 v2.4.2, Picard tools v2.21.2<br>Differential expression analysis: featureCounts v1.5.3, DESeq2 v1.26.0<br>Genome browser tracks: deepTools v3.3.2 or v3.5.0<br>Peak calling and analysis: MACS2 v2.2.7.1, DiffBind v3.8.4<br>Structure modelling: UCSF Chimera v1.16, ZDOCK v3.0.2<br>Statistical analysis and visualisation: R v3.6.2, pheatmap v1.0.12, sinkr v0.6, eulerr package (v7.0.2)<br>Image analysis: Fiji/ImageJ software v1.54p |

For manuscripts utilizing custom algorithms or software that are central to the research but not yet described in published literature, software must be made available to editors and reviewers. We strongly encourage code deposition in a community repository (e.g. GitHub). See the Nature Portfolio guidelines for submitting code & software for further information.

## Data

Policy information about availability of data

All manuscripts must include a data availability statement. This statement should provide the following information, where applicable:

- Accession codes, unique identifiers, or web links for publicly available datasets
- A description of any restrictions on data availability
- For clinical datasets or third party data, please ensure that the statement adheres to our policy

Sequencing data generated in this study has been deposited to the GEO under accession (GSE247156). The following data were retrieved from GEO: RNA-seq, sRNA-seq and ChIP-seq data for Rhi and Kipf were downloaded from the GEO (accession GSE202468),. HP1a ChIP-seq data was downloaded from GEO (GSE140542). Rhi ChIP-seq samples from control, Rhi, and Moon knockout ovaries input samples were downloaded from GEO (accession GSE97719), sRNA-seq data for Drosophila species (GSE225888). The dm6 genome assembly was downloaded from the UCSC genome browser and the following assemblies were downloaded from RefSeq: GCF_003285975, GCF_003286155, GCF_016746395, and GCF_016746365. Source data are provided with this paper.

## Research involving human participants, their data, or biological material

Policy information about studies with human participants or human data. See also policy information about sex, gender (identity/presentation), and sexual orientation and race, ethnicity and racism.

| | |
|---|---|
| Reporting on sex and gender | N/A |
| Reporting on race, ethnicity, or other socially relevant groupings | N/A |
| Population characteristics | N/A |
| Recruitment | N/A |
| Ethics oversight | N/A |

Note that full information on the approval of the study protocol must also be provided in the manuscript.

# Field-specific reporting

Please select the one below that is the best fit for your research. If you are not sure, read the appropriate sections before making your selection.

☒ Life sciences ☐ Behavioural & social sciences ☐ Ecological, evolutionary & environmental sciences

For a reference copy of the document with all sections, see nature.com/documents/nr-reporting-summary-flat.pdf

# Life sciences study design

All studies must disclose on these points even when the disclosure is negative.

| | |
|---|---|
| Sample size | As the number of flies to be used in the experiments was not a limiting factor, no statistical power analyses were used to predetermine sample sizes. Sample sizes were chosen as large as possible while still practically feasible in term of data collection. All experiments in this work build on established experimental schemes in the piRNA and Drosophila genetics fields. Adequate statistics has been applied throughout the manuscript in order to make sure that the observed effects are significant given the reported sample size. |
| Data exclusions | Two CUT&RUN samples were excluded from the analysis due to failing quality control, as stated in the manuscript. These samples are marked as failed but nevertheless made available. |
| Replication | Experiments were independently repeated, the number of replicates are presented in the figure legends and/or methods. |
| Randomization | Randomization was not relevant to this study. All flies were from similar genetic background and groups for statistical comparisons were constructed based on treatment (knockdown target). |
| Blinding | No investigator blinding was applied during data acquisition or analyses as the data was mostly analyzed in bulk by (blind) scripts such as for NGS or FISH quantification or blinding was not desirable for data presentation. |

# Reporting for specific materials, systems and methods

We require information from authors about some types of materials, experimental systems and methods used in many studies. Here, indicate whether each material, system or method listed is relevant to your study. If you are not sure if a list item applies to your research, read the appropriate section before selecting a response.

## Materials & experimental systems

| n/a | Involved in the study |
|-----|-----------------------|
| ☐ | ☒ Antibodies |
| ☐ | ☒ Eukaryotic cell lines |
| ☒ | ☐ Palaeontology and archaeology |
| ☐ | ☒ Animals and other organisms |
| ☒ | ☐ Clinical data |
| ☒ | ☐ Dual use research of concern |
| ☒ | ☐ Plants |

## Methods

| n/a | Involved in the study |
|-----|-----------------------|
| ☐ | ☒ ChIP-seq |
| ☒ | ☐ Flow cytometry |
| ☒ | ☐ MRI-based neuroimaging |

## Antibodies

| | |
|---|---|
| Antibodies used | anti-Rhino polyclonal, Rabbit pAB ,Eurogentec, Mohn et al., 2014, ChIP: 5µl; IF 1:1000<br>anti-Histone H3K9me3 antibody, Rabbit pAB , Active motif #39161, ChIP: 5µl; CUT&RUN 1:20<br>anti-Histone H3K27me3 antibody, Rabbit mAB, Cell Signaling Technology  #9733S, CUT&RUN 1:20<br>anti-Histone H3K27me3 antibody, Rabbit pAB, Millipore #07-449, ChIP 5µl<br>anti-H2AK119Ub antibody, Rabbit mAB, Cell Signaling Technology  #8240, CUT&RUN 1:20<br>anti-FLAG M2 antibody, Mouse mAB, Sigma Aldrich #F1804, CUT&RUN 1:20; WB: 1:2500<br>anti-Kipf A96 antibody, Mouse mAB, Baumgartner et al., 2022, Brennecke lab, WB: 1:200<br>anti-alphaTubulin antibody, Rabbit pAB, Abcam #ab18251, WB: 1:5000<br>anti-Rabbit IgG IRDye 680LT, Goat pAB, Licor #926-68021, WB 1:10000<br>anti-Mouse IgG IRDye 800CW, Goat pAB, Licor #926-32210, WB 1:5000 |
| Validation | All commercially available antibodies were validated by the manufacturers. Non-commercially available antibodies used in this study have been validated in previous publications (cited in this manuscript).<br><br>All validation statements, including citation for commercial antibodies can be found on the manufacturers' websites:<br>anti-Histone H3K9me3: https://www.activemotif.com/catalog/details/39161/histone-h3-trimethyl-lys9-antibody-pab<br>anti-Histone H3K27me3 (Cell signalling):: https://www.cellsignal.com/products/primary-antibodies/tri-methyl-histone-h3-lys27-c36b11-rabbit-mab/9733<br>anti-Hitsone H3K27me3 (Milipore):https://www.merckmilllipore.com/DE/en/product/Anti-trimethyl-Histone-H3-Lys27-Antibody,MM_NF-07-449<br>anti-H2AK119Ub: https://www.cellsignal.com/products/primary-antibodies/ubiquityl-histone-h2a-lys119-d27c4-xp-rabbit-mab/8240<br>anti-FLAG M2: https://www.sigmaaldrich.com/DE/en/product/sigma/f1804<br>anti-Kipf : The antoby was validated  in Baugmarten et al., 2022<br>anti-aplhaTubulin: https://www.abcam.com/en-us/products/primary-antibodies/alpha-tubulin-antibody-microtubule-marker-ab18251 |

## Eukaryotic cell lines

Policy information about cell lines and Sex and Gender in Research

| | |
|---|---|
| Cell line source(s) | Used cell line in this study: Drosophila melanogaster Schneider 2 cells (S2 cells). S2 cells were purchased from ThermoFisher (#R69007). |
| Authentication | Drosophila S2 cells were not authenticated, however, next-generation sequencing confirmed their identify as Drosophila cells. |
| Mycoplasma contamination | S2 cells were regularly checked for mycoplasma by an in-house facility. The cell line used was mycoplasma negative. |
| Commonly misidentified lines<br>(See ICLAC register) | N/A |

## Animals and other research organisms

Policy information about studies involving animals; ARRIVE guidelines recommended for reporting animal research, and Sex and Gender in Research

| | |
|---|---|
| Laboratory animals | This study exclusively involved work with Drosophila melanogaster, a standard invertebrate model organism that does not underlie any ethical restrictions. Standard laboratory procedures have been applied throughout the study. Drosophila strains used in this study are described in the Methods section. Adult females used for ovary dissections were aged 2–6 days as stated in the Methods section. |
| Wild animals | The study did not involve wild animals. |
| Reporting on sex | The study focused on the ovarian function of E(z) and Rhino and therefore used only females. |

| Field-collected samples | The study did not involve samples collected from the wild. |
|---|---|
| Ethics oversight | No ethical approval or guidance were required as the study used Drosophila. |

Note that full information on the approval of the study protocol must also be provided in the manuscript.

## Plants

| Seed stocks | N/A |
|---|---|
| Novel plant genotypes | N/A |
| Authentication | N/A |

## ChIP-seq

### Data deposition

☒ Confirm that both raw and final processed data have been deposited in a public database such as GEO.

☒ Confirm that you have deposited or provided access to graph files (e.g. BED files) for the called peaks.

| Data access links *May remain private before publication.* | https://www.ncbi.nlm.nih.gov/geo/query/acc.cgi?acc=GSE247156 |
|---|---|
| Files in database submission | H3K27me3.nos-E(z).1 [ChIP-seq]<br>H3K27me3.nos-E(z).2 [ChIP-seq]<br>H3K27me3.nos-E(z).3 [ChIP-seq]<br>H3K27me3.nos-white.1 [ChIP-seq]<br>H3K27me3.nos-white.2 [ChIP-seq]<br>H3K27me3.nos-white.3 [ChIP-seq]<br>H3K9me3.nos-E(z).1 [ChIP-seq]<br>H3K9me3.nos-E(z).2 [ChIP-seq]<br>H3K9me3.nos-white.1 [ChIP-seq]<br>H3K9me3.nos-white.2 [ChIP-seq]<br>input.nos-E(z).1 [ChIP-seq]<br>input.nos-E(z).2 [ChIP-seq]<br>input.nos-white.1 [ChIP-seq]<br>input.nos-white.2 [ChIP-seq]<br>Rhino.nos-E(z).1 [ChIP-seq]<br>Rhino.nos-E(z).2 [ChIP-seq]<br>Rhino.nos-white.1 [ChIP-seq]<br>Rhino.nos-white.2 [ChIP-seq]<br>H3K27me3.nos-Kipf.1.2 [ChIP-seq]<br>H3K27me3.nos-Kipf.2.2 [ChIP-seq]<br>H3K27me3.nos-White.1.2 [ChIP-seq]<br>H3K27me3.nos-White.2.2 [ChIP-seq]<br>H3K27me3.nos-White.3.2 [ChIP-seq]<br>input.nos-Kipf.1.2 [ChIP-seq]<br>input.nos-White.1.2 [ChIP-seq] |
| Genome browser session (e.g. UCSC) | UCSC Genome Browser-compatible BigWig (.bw) files are included in the GEO submission. |

### Methodology

| Replicates | 2-3 biological replicates were performed per sample. |
|---|---|
| Sequencing depth | Name Total Unique Length Type<br>H3K27me3.nos-E(z).1 14556897.00  2861898.00  100 single<br>H3K27me3.nos-E(z).2 32064702.00  20754163.00  100 single<br>H3K27me3.nos-E(z).3 35148363.00  22601795.00  100 single<br>H3K27me3.nos-white.1 16004969.00  3838357.00  100 single<br>H3K27me3.nos-white.2 21280635.00  10288146.00  100 single<br>H3K27me3.nos-white.3 31984551.00  23322089.00  100 single<br>H3K9me3.nos-E(z).1 36960490.00  11622760.00  100 paired-end |

H3K9me3.nos-E(z).2 30653102.00  11795243.00  100 paired-end
H3K9me3.nos-white.1 36524968.00  12982482.00  100 paired-end
H3K9me3.nos-white.2 39929034.00  11551417.00  100 paired-end
input.nos-E(z).1 20141957.00  17048831.00  100 single
input.nos-E(z).2 32559246.00  27390817.00  100 paired-end
input.nos-white.1 20323312.00  16403946.00  100 single
input.nos-white.2 31386746.00  26555400.00  100 paired-end
Rhino.nos-E(z).1 39255318.00  11981984.00  100 paired-end
Rhino.nos-E(z).2 36020694.00  19117573.00  100 paired-end
Rhino.nos-white.1 38668340.00  28323873.00  100 paired-end
Rhino.nos-white.2 47157362.00  28907720.00  100 paired-end
H3K27me3_Wh.1.2 83310948.00  50015531.00  100 paired-end
H3K27me3_Wh.2.2 95069608.00  55097575.00  100 paired-end
H3K27me3_Wh.3.2 146792032.00  81935056.00  100 paired-end
H3K27me3_Kip.1.2 71411850.00  40050953.00  100 paired-end
H3K27me3_Kip.2.2 83203520.00  46435848.00  100 paired-end
input_White.1.2 89380548.00  73077128.00  100 paired-end
Input_Kipferl.1.2 130128714.00  106785429.00  100 paired-end

**Antibodies**

ChIPseq_anti Rhino samples: Anti-Rhino polyclonal antibody produced in Rabbit (Mohn et al 2014)
ChIPseq_anti H3K27me3 samples: anti-H3K27me3 (Millipore 07-449)
ChIPseq_anti H3K9me3 samples: anti-H3K9me3 (active motif 39161)

**Peak calling parameters**

Peaks were called using MACS2 (v2.2.7.1) to capture narrow (-q 0.05 -g dm) and broad peaks (-q 0.05 -g dm —broad —broad-cutoff 0.1). Only uniquely mapped reads were used for the peak calling. As a control we used Input for each condition.

**Data quality**

Visual inspection of data in the genome browser to confirm that the ChIPseq signal in wildtype accumulates as expected based on previous literature

**Software**

ChIP-seq reads were aligned to the dm6 genome using Bowtie2 (v2.4.2), with the alignment process set to report at most one hit for each read. In case of alignments with the same MAPQ score, the best alignment was randomly selected from among those equally scored alignments. Peaks were called using MACS2 (v2.2.7.1) to capture narrow (-q 0.05 -g dm) and broad peaks (-q 0.05 -g dm --broad --broad-cutoff 0.1). Only uniquely mapped reads were used for the peak calling. As a control we used Input for each condition.

