## [Peer Review File · Nature Structural & Molecular Biology]

Binding of heterochromatin protein Rhino to a subset of piRNA clusters depends on a combination of two histone marks

Corresponding Author: Professor Gregory Hannon

Version 0:

Decision Letter:

21st Dec 2023

Dear Dr. Hannon,

Thank you again for submitting your manuscript "A dual histone code specifies the binding of heterochromatin protein Rhino to a subset of piRNA source loci".

As per our last email, we are still waiting for you to submit your manuscript checklists and any links to data needed for peer review, before we are able to initiate the peer review process.

Since we received your out-of-office reply, and are, ourselves, operating at minimum capacity over the holidays, I am re-opening the manuscript submission link for you to resubmit your manuscript with all the associated files needed for the peer review process directly to our system, at your convenience:

Link Redacted

Sincerely, and happy holidays,
Sara

Sara Osman, Ph.D.
Associate Editor
Nature Structural & Molecular Biology

Version 1:

Decision Letter:

22nd Feb 2024

Dear Dr. Hannon,

Thank you again for submitting your manuscript "A dual histone code specifies the binding of heterochromatin protein Rhino to a subset of piRNA source loci". I apologize for the delay in responding, which resulted from the difficulty in obtaining suitable referee reports. Nevertheless, we now have comments (below) from the 2 reviewers who evaluated your paper. In light of those reports, we remain interested in your study and would like to see your response to the comments of the referees, in the form of a revised manuscript.

Please be sure to address/respond to all concerns of the referees in full in a point-by-point response and highlight all changes in the revised manuscript text file. If you have comments that are intended for editors only, please include those in a

separate cover letter.

We expect to see your revised manuscript within 3-6 months. If you cannot send it within this time, please contact us to discuss an extension; we would still consider your revision, provided that no similar work has been accepted for publication at NSMB or published elsewhere.

Reporting Summary:

Please note that all key data shown in the main figures as cropped gels or blots should be presented in uncropped form, with molecular weight markers. These data can be aggregated into a single supplementary figure item. While these data can be displayed in a relatively informal style, they must refer back to the relevant figures. These data should be submitted with the final revision, as source data, prior to acceptance, but you may want to start putting it together at this point.

SOURCE DATA: we request that authors provide, in tabular form, the data underlying the graphical representations used in figures. This is to further increase transparency in data reporting, as detailed in this editorial (<http://www.nature.com/nsmb/journal/v22/n10/full/nsmb.3110.html>). Spreadsheets can be submitted in excel format. Only one (1) file per figure is permitted; thus, for multi-paneled figures, the source data for each panel should be clearly labeled in the Excel file; alternately the data can be provided as multiple, clearly labeled sheets in an Excel file. When submitting files, the title field should indicate which figure the source data pertains to. We encourage our authors to provide source data at the revision stage, so that they are part of the peer-review process.

Data availability: this journal strongly supports public availability of data. All data used in accepted papers should be available via a public data repository, or alternatively, as Supplementary Information. If data can only be shared on request, please explain why in your Data Availability Statement, and also in the correspondence with your editor. Please note that for some data types, deposition in a public repository is mandatory - more information on our data deposition policies and available repositories can be found below:

<https://www.nature.com/nature-research/editorial-policies/reporting-standards#availability-of-data>

We require deposition of coordinates (and, in the case of crystal structures, structure factors) into the Protein Data Bank with the designation of immediate release upon publication (HPUB). Electron microscopy-derived density maps and coordinate data must be deposited in EMDDB and released upon publication. Deposition and immediate release of NMR chemical shift assignments are highly encouraged. Deposition of deep sequencing and microarray data is mandatory, and the datasets must be released prior to or upon publication. To avoid delays in publication, dataset accession numbers must be supplied with the final accepted manuscript and appropriate release dates must be indicated at the galley proof stage.

Nature Structural & Molecular Biology is committed to improving transparency in authorship. As part of our efforts in this direction, we are now requesting that all authors identified as 'corresponding author' on published papers create and link their Open Researcher and Contributor Identifier (ORCID) with their account on the Manuscript Tracking System (MTS), prior to acceptance. This applies to primary research papers only. ORCID helps the scientific community achieve unambiguous attribution of all scholarly contributions. You can create and link your ORCID from the home page of the MTS by clicking on 'Modify my Springer Nature account'. For more information please visit please visit

<http://www.springernature.com/orcid>>www.springernature.com/orcid.

Link Redacted

Sincerely,
Sara

Sara Osman, Ph.D.
Associate Editor
Nature Structural & Molecular Biology

Referee expertise:

Referee #1: heterochromatin, piRNAs, drosophila

Referee #2: Transcription regulation

Reviewers' Comments:

Reviewer #1:

Remarks to the Author:

Review of NSMB-A48537A by Akkouche, Kneuss, Barnelöv and colleagues.

Eukaryotes universally generate small regulatory RNAs to suppress foreign genetic material. In the animal germline, the dominant system for transposon suppression is the piRNA pathway. PiRNAs are produced from long, single-stranded precursors transcribed from specific genomic piRNA source loci, often referred to as piRNA clusters. In the *Drosophila* germline, the major class of piRNA source loci is embedded in heterochromatin marked by H3K9me3. These loci are transcribed on both genomic strands in a non-canonical manner, and this process depends on the presence of the HP1 protein Rhino, which brings the necessary protein machinery to chromatin. The targeting of Rhino to only a subset of heterochromatic loci has been a persistent question in the field. The authors of this manuscript shed light on this mystery by identifying a molecular principle, based on two different histone marks, that significantly contributes to Rhino's chromatin binding specificity.

All in all, this is a great manuscript that is well placed in a visible journal such as NCB. The authors combine a broad range of complementary experiments and approaches to come to a compelling model that the K9me3 and the K27me3 marks synergize at an important subset of Rhino loci genome wide. As carefully dissected by the authors, this mechanism is a second Rhino recruitment mode, independent from the recently discovered mode that depends on the Zinc-finger protein Kipferl.

I support publication of this work but have two suggestions for the authors to consider and a few minor comments that are easy to implement (listed below).

- The elephant in the room is that a direct demonstration at the biochemical level that supports the compelling and in principle simple model is missing. As stated in the last sentence of the discussion by the authors, previous work has failed to measure an affinity of the recombinant Rhino domain to H3K27me3 peptides. Moreover, the dimerization of the Rhino chromo-domain has received surprisingly conflicting conclusions in the field. While dimerization has been shown in the Rhino crystal structure paper (Yu et al 2015), our own recent work has surprisingly failed to detect Rhino chromo-domain dimerization in solution (Baumgartner et al. 2023). My point here is the following: The biochemical interrogation of the proposed model does not strike me as too difficult as long as an H3K9me3, K27me3 peptide can be synthesized or purchased. The result of this experiment would not jeopardize the paper in any way, it would just make it more complete. If supportive, great. If not, then there is something really interesting out there that must underlie the beautiful observations. I leave it up to the authors to provide this experiment. But if they decide against it, there should be a more visible section in the discussion clearly pointing out the possibility of an alternative mechanism. Speaking of dimerization, if the Rhino CD does indeed dimerize, then would a single Rhino CD in the Tuncay-style experiments in S2-cells not be sufficient? And finally, a mutation in the fly where the dimerization capacity of the CD is affected but Kipferl-binding as well as K9me3 binding is unaffected would be super interesting. Have the authors considered making such a fly?

- A second interesting addition to the work would be a double depletion of E(z) and Kipferl. Again, an experiment that would elevate the paper irrespective of the outcome. Especially in light of the Satellite regions that bind more Rhino in Kipferl

depleted ovaries. Are these now E(z) dependent or not? Is there potentially a third mechanism how Rhino can be recruited?

Minor points:

- Transposon or TE? Make consistent throughout the text.
- The last two sentences of the abstract are not very strong as an ending in my opinion. Consider revising.
- While there are no guidelines in the field, it might make sense to use piRNA source loci throughout the manuscript. The current text is in my opinion confusing in terms of what is a cluster and what is a source locus. Clusters are also piRNA source loci in my opinion.
- First paragraph of the result section: I am not sure whether that is the best motivation for the work. There is a bigger, more exciting question behind.
- The second half of the 2nd paragraph of the result section reads like a discussion paragraph. Consider shifting there.
- Legend for Figure 1: the males do not express an shRNA construct.
- Figure 1 analysis: are the 1kb tiles and the mapping data taking only genome-unique mappers into account? If so, please state this clearly.
- Cluster 20A and flamenco are often lumped together as control for uni-strand clusters. While that is true, the other important distinction is that flamenco is somatic (where Rhino is not expressed) and 20A predominantly germline. It would help to make this clear to the reader once.
- On the genetic mini-screen: how do the authors explain that the different PRC2 complex members show such different results in the TE de-repression? Burdock compared to HeT-A for example. A small statement on the author's thinking here would be appreciated in the figure legend.
- Add statistics to Figure S2e and other panels where possible.
- Figure 4i is very difficult to read as there are so many colors and dots. Consider a more accessible panel layout.

Julius Brennecke

Reviewer #2:

Remarks to the Author:

Rhi binding to H3K9me3-marked sites to turn on piRNA expression, which subsequently targets TE for degradation or silencing. The recruitment of Rhi to target sites is only partially explained by Kipf. In the manuscript, the authors first reported that the knockdown of Ez and several other PRC2 subunits in germ cells resulted in the derepression of transposons, suggesting a critical role of PRC2 and H3K27me3 on TE silencing through a partial contribution to Rhi function. They showed the Ez knockdown reduced 42AB piRNA, but not 20A and 80F clusters. Genome-wide analyses showed that H3K27me3 co-occupied with H3K9me3 and Rhi at 42AB loci. Overall, almost all Rhi-bound sites are also bound by H3K9me3, about half of which also contain H3K27me3 mark. The authors further showed that Rhi binding to some piRNA source loci is dependent on Ez, through Ez-knockdown assays. However, control experiments were not included to show that they are indeed independent of Kipf. Some of the key findings were further confirmed using S2 cells with the expression of synthetic dual-chromatin reader domains.

The manuscript is well-written, and the results are clearly described. The quality of the data presented is great. The findings that Ez is required for some piRNA function and TE expression and the presence of dual H3K9me3 and H3K27me3 at some piRNA source loci are novel and interesting. However, the experiments are limited in scope. Some essential control experiments that would have been very helpful in discerning the two mechanisms of Rhi recruitment to target loci were not comparatively done. As such, the mechanism remains elusive. It seems H3K9me3 is also present at Ez-dependent Rhi binding sites, but their role is unclear.

Specific comments:

Fig.1: is H3K27me3 required for this role of Ez? Can an Ez methylation mutant rescue the phenotype?

Fig.2c. Can the authors show genome-wide distribution and overlap between Ez and Kipf? This is necessary to support the key conclusion that there are distinct Kipf- and Ez-mediated Rhi bindings.

Fig.3. To support the conclusion that Ez and Kipf may provide two distinct modes of Rhi recruitment to separate sites, can the authors show how the knockdown of Kipf or Ez, respectively, affect Rhi binding at piRNA source loci? Also, will they affect H3K9me3 and/or H3K27me3 binding at these sites? Presumably, Ez depletion will surely affect H3K27me3.

Fig.3e: what's the level of H3K9me3 and H3K27me3 on these sites in control and Ez-knockdown cells? Is the reduction of Rhi binding at 42AB caused by H3K27me3 loss in the knockdown cells?

Fig.3g: the information provided here is fragmented. Can the authors show H3K9me3 and H3K27me3 in control and kipf-knockdown cells in comparison to control and Ez-knockdown cells?

Fig.4: similar to comments in Fig.3 above, the experimental design missed important controls to allow full comparison of the two categories of Ez- or kipf-dependent sites. For instance, 4c should include k27, 4d should include k9, and so forth.

Fig.5b: There is a strong correlation between H3K9me3 level, Rhi ChIP-seq signal, and piRNA expression. In contrast, the H3K27me3 level does not seem to be correlated with any of the above. In the site with low H3K9me3 (left of the curves), there were some bins with high H3K27me3. However, Rhi binding and piRNA expression were not increased in these bins.

Version 2:

Decision Letter:

Our ref: NSMB-A48537B

19th Dec 2024

Dear Dr. Hannon,

Thank you for submitting your revised manuscript "A dual histone code specifies the binding of heterochromatin protein Rhino to a subset of piRNA source loci" (NSMB-A48537B). It has now been seen by the original referees and their comments are below. The reviewers find that the paper has improved in revision, and therefore we'll be happy in principle to publish it in Nature Structural & Molecular Biology, pending minor revisions to satisfy the referees' final requests and to comply with our editorial and formatting guidelines.

We are now performing detailed checks on your paper and will send you a checklist detailing our editorial and formatting requirements within the next few weeks. Please note that some delays are to be expected due to the holidays at this time of the year. Please do not upload the final materials and make any revisions until you receive this additional information from us.

To facilitate our work at this stage, it is important that we have a copy of the main text as a word file. If you could please send along a word version of this file as soon as possible, we would greatly appreciate it; please make sure to copy the NSMB account (cc'ed above).

Thank you again for your interest in Nature Structural & Molecular Biology. Please do not hesitate to contact me if you have any questions.

Sincerely,
Sara

Sara Osman, Ph.D.
Senior Editor
Nature Structural & Molecular Biology

Reviewer #1 (Remarks to the Author):

I apologise for the delay in reviewing the revised manuscript. The authors have, in my opinion, responded adequately to the points raised by both reviewers. The manuscript and the data presented are of high quality, the text and discussion are balanced, outline the new findings in the context of the field and point out the limitations of the work presented. Overall, this is an important paper that will stimulate further research in this exciting area of chromatin biology.

Reviewer #2 (Remarks to the Author):

The authors have addressed all of my concerns.

Version 3:

Decision Letter:

2nd May 2025

Dear Dr. Hannon,

We are now happy to accept your revised paper "Binding of heterochromatin protein Rhino to a subset of piRNA clusters depends on a combination of two histone marks" for publication as an article in Nature Structural & Molecular Biology.

Your paper will be published online soon after we receive proof corrections and will appear in print in the next available issue. You can find out your date of online publication by contacting the production team shortly after sending your proof corrections.

Authors may need to take specific actions to achieve <https://www.springernature.com/gp/open-research/funding/policy-compliance-faqs> compliance with funder and institutional open access mandates. If your research is supported by a funder that requires immediate open access (e.g. according to <https://www.springernature.com/gp/open-research/plan-s-compliance> Plan S principles) then you should select the gold OA route, and we will direct you to the compliant route where possible. For authors selecting the subscription publication route, the journal's standard licensing terms will need to be accepted, including <https://www.springernature.com/gp/open-research/policies/journal-policies> self-archiving policies. Those licensing terms will supersede any other terms that the author or any third party may assert apply to any version of the manuscript.

Sincerely,
Sara

Sara Osman, Ph.D.
Senior Editor
Nature Structural & Molecular Biology

Response to referees

We are very grateful to the two reviewers for their valuable feedback and suggestions. We performed new experiments and analyses and made changes to the text of the paper to address the comments. Below, we provide our point-by-point response (shown in blue) to the comments of each reviewer.

In addition to these requested changes, we also repeated the chromodomain experiments presented in **Fig. 4** and **Extended Data Figures 6-7**. This was prompted by finding a non-synonymous mutation in the CBX1 chromodomain of our previously used dual binder CD^{Pc-CBX1} construct. To avoid introducing batch effects, we therefore repeated all chromodomain experiments (CD^{Pc-CBX1}, CD^{2xRhi}, CD^{2xCBX1}, CD^{2xPc}) in control, *E(z)* and *Kipf* knockdown conditions. Our original conclusions remain unchanged as no qualitative differences were observed compared to the previous analyses, but we have replaced the corresponding figure panels with new ones showing the new experiments only (**Fig. 4** and **Extended Data Figure 6-7**).

Reviewer #1:

Remarks to the Author:

Review of NSMB-A48537A by Akkouche, Kneuss, Bernelöv and colleagues.

Eukaryotes universally generate small regulatory RNAs to suppress foreign genetic material. In the animal germline, the dominant system for transposon suppression is the piRNA pathway. PiRNAs are produced from long, single-stranded precursors transcribed from specific genomic piRNA source loci, often referred to as piRNA clusters. In the *Drosophila* germline, the major class of piRNA source loci is embedded in heterochromatin marked by H3K9me3. These loci are transcribed on both genomic strands in a non-canonical manner, and this process depends on the presence of the HP1 protein Rhino, which brings the necessary protein machinery to chromatin. The targeting of Rhino to only a subset of heterochromatic loci has been a persistent question in the field. The authors of this manuscript shed light on this mystery by identifying a molecular principle, based on two different histone marks, that significantly contributes to Rhino's chromatin binding specificity.

All in all, this is a great manuscript that is well placed in a visible journal such as NCB. The authors combine a broad range of complementary experiments and approaches to come to a compelling model that the K9me3 and the K27me3 marks synergize at an important subset of Rhino loci genome wide. As carefully dissected by the authors, this mechanism is a second Rhino recruitment mode, independent from the recently discovered mode that depends on the Zinc-finger protein Kipferl.

I support publication of this work but have two suggestions for the authors to consider and a few minor comments that are easy to implement (listed below).

We thank the referee for their summary of our contribution and for their support and constructive feedback.

- The elephant in the room is that a direct demonstration at the biochemical level that supports the compelling and in principle simple model is missing. As stated in the last sentence of the discussion by the authors, previous work has failed to measure an affinity of the recombinant Rhino domain to H3K27me3 peptides. Moreover, the dimerization of the Rhino chromo-domain has received surprisingly conflicting conclusions in the field. While dimerization has been shown in the Rhino crystal structure paper (Yu et al 2015), our own recent work has surprisingly failed to detect Rhino chromo-domain dimerization in solution (Baumgartner et al. 2023). My point here is the following: The biochemical interrogation of the proposed model does not strike me as too difficult as long as an H3K9me3, K27me3 peptide can be synthesized or purchased. The result of this experiment would not jeopardize the paper in any way, it would just make it more complete. If supportive, great. If not, then there is something really interesting out there that must underlie the beautiful observations. I leave it up to the authors to provide this experiment. But if they decide against it, there should be a more visible section in the discussion clearly pointing out the possibility of an alternative mechanism.

We agree that biochemically demonstrating Rhino dimerization and simultaneous binding to H3K9me3 and H3K27me3 would provide strong additional evidence for the proposed model. Unfortunately, we have not been able to get the biochemical characterization off the ground within the provided timeframe. Nevertheless, we have included additional bioinformatic analyses (**Extended Data Fig. 6g,h**) showing that the CD^{2xRhi} is capable of binding both CD^{2xCBX1} and CD^{2xPc} targets in S2 cells (that is, regions

primarily enriched in H3K9me3 and H3K27me3, respectively). Moreover, we have performed additional control experiments, showing that neither CD^{2x}CBX1, CD^{2x}Pc, CD^{Pc}-CBX1, nor CD^{2x}Rhi display any major alterations in their binding pattern upon Kipf depletion in S2 cells, providing additional confirmation that the association with H3K9-K27me3-decorated regions is Kipf-independent (**Extended Data Fig. S6k-n**).

Considering this and in response to the reviewer's feedback, we have toned-down our claims about Rhi CD dimerization and extended our discussion regarding alternative mechanisms that may govern Rhi recruitment. Overall, we think we have strengthened the revised manuscript and that our study has shed new light on important aspects of Rhi recruitment and chromatin readers.

Speaking of dimerization, if the Rhino CD does indeed dimerize, then would a single Rhino CD in the Tuncay-style experiments in S2-cells not be sufficient?

We have attempted to profile the binding of a single Rhi chromodomain (CD^{1x}Rhi) using CUT&RUN. However, the CD^{1x}Rhi construct did not provide any enrichment above background, despite the construct showing robust expression in S2 cells. While this could indicate that the Rhi CD does not dimerize in vivo, this may also be due to insufficient binding affinity of a single CD, as reported elsewhere (Villasenor et al. 2020), complicating our interpretation of the results. We have included these data in **Extended Data Fig. 8d** and extended the discussion to put our results in light of these new results as well as the ones reported recently by Baumgartner and colleagues (Baumgartner et al. 2024).

And finally, a mutation in the fly where the dimerization capacity of the CD is affected but Kipferl-binding as well as K9me3 binding is unaffected would be super interesting. Have the authors considered making such a fly?

We agree that the suggested experiment would potentially help elucidate the role of Rhi dimerization, however, we believe given the time and resources required for obtaining and studying mutant flies – if possible at all – this is beyond the scope of our study.

- A second interesting addition to the work would be a double depletion of E(z) and Kipferl. Again, an experiment that would elevate the paper irrespective of the outcome. Especially in light of the Satellite regions that bind more Rhino in Kipferl depleted ovaries. Are these now E(z) dependent or not? Is there potentially a third mechanism how Rhino can be recruited?

We thank the reviewer for raising several interesting questions. To answer these, we first investigated the expression of satellite regions (1.688 and Rsp) using RNA-FISH following germline-specific knockdown of E(z). Our results revealed a strong and significant loss of signal for both 1.688 and Rsp, while a pronounced increase in RNA FISH signal was observed upon Kipf knockdown, consistent with findings from (Baumgartner et al. 2022) (**Fig. R1**). These results suggest that the expression of Kipf-independent 1.688 and Rsp satellite regions instead may depend on E(z).

Fig. R1: Confocal images showing RNA-FISH for the indicated satellite regions in control, E(z) knockdown, and kipf knockdown as well as in E(z) and kipf double-knockdown, using the nos-Gal4 driver (scale bar: 5 µm).

Additionally, we sequenced piRNA populations from *Drosophila* ovaries with germline-specific depletion of E(z) and Kipf, comparing them to control knockdowns (targeting *w*) and single depletions of E(z) or Kipf (Fig. R2). Our analysis revealed that E(z) depletion results in the reduction of piRNAs from *Rsp* repeats while piRNA production varied across 1.688 satellite regions, with different regions showing different responses (Fig. R2a). In contrast, both 1.688 and *Rsp* piRNAs were strongly upregulated in the double-KD of E(z) and Kipf, similar to the results observed in the single Kipf knockdown (Fig. R2b,d).

Given these opposing results, it is challenging to draw a clear conclusion from all the experiments conducted on the satellite regions. The single KD experiments, suggest that these regions are E(z)-dependent based on RNA-FISH and piRNA production (Fig. R1, Fig. R2). However, in the double-KD experiments, piRNA production indicates that they are independent of both E(z) and Kipf, as piRNA levels were higher in the absence of both factors compared to ctrl (Fig. R2b).

Given that we can only speculate about the differences between the RNA-FISH experiment and the sRNA-seq results for the double KD, we feel that discussing these new data would confuse the main message of our work. We therefore propose to not include these results in the manuscript, with the hope that we would understand this interesting set of observations much more clearly in the future.

Fig. R2: Satellite sRNA-seq. Scatter plots depicting normalized piRNA levels of all piRNAs mapping in 1kb bins in ovaries with the indicated knockdowns (average of one or two replicates each). PiRNAs mapping to specific piRNA clusters or satellites regions are indicated (see legend in panel a).

Minor points:

- Transposon or TE? Make consistent throughout the text.

We have unified the text and now use 'TE' instead of 'transposon' throughout the manuscript.

- The last two sentences of the abstract are not very strong as an ending in my opinion. Consider revising.

We thank the reviewer for this suggestion and revised the end of the abstract to provide a stronger conclusion.

- While there are no guidelines in the field, it might make sense to use piRNA source loci throughout the manuscript. The current text is in my opinion confusing in terms of what is a cluster and what is a source locus. Clusters are also piRNA source loci in my opinion.

In agreement with the reviewer, we consider piRNA clusters a subset of the wider definition "piRNA source loci" which also include genic piRNAs or satellite repeats. We added text in the introduction to make these distinctions (and those of unistrand and dual-strand piRNA clusters) clearer to the general audience.

- First paragraph of the result section: I am not sure whether that is the best motivation for the work. There is a bigger, more exciting question behind.

We agree with the reviewer's comment. We have revised the punchline of the first paragraph accordingly.

- The second half of the 2nd paragraph of the result section reads like a discussion paragraph. Consider shifting there.

The reviewer is right, we have moved this part to the discussion section.

- Legend for Figure 1: the males do not express an shRNA construct.

We thank the reviewer for pointing this out. We have changed the legend accordingly and replaced “expressing” with “carrying”.

- Figure 1 analysis: are the 1kb tiles and the mapping data taking only genome-unique mappers into account? If so, please state this clearly.

This was specified in the methods section but not in the figure legend. We have now added the statement (uniquely mapped) to the legends of **Fig. 1e-h**.

Please note that **Fig. 1e-f** have been updated as the previous version used spliced alignment to calculate the mappability, increasing the number of bins considered to be mappable. The mappability is now calculated using bowtie.

- Cluster 20A and flamenco are often lumped together as control for uni-strand clusters. While that is true, the other important distinction is that flamenco is somatic (where Rhino is not expressed) and 20A predominantly germline. It would help to make this clear to the reader once.

We have clarified this distinction in the text the first time we cited *flam* and *20A*.

- On the genetic mini-screen: how do the authors explain that the different PRC2 complex members show such different results in the TE de-repression? Burdock compared to HeT-A for example. A small statement on the author’s thinking here would be appreciated in the figure legend.

We have added a supplementary note (**Supplementary note 1**) addressing the disparities observed between knockdowns of different PRC2 complex members:

Supplementary note 1

Depletion of *caf1-55* resulted in rudimentary ovaries rendering the analyses of the germline TE expression difficult. Knockdowns of *esc*, *esc1* and *Su(z)12* did not result in sterility of the F1 generation contrary to expected; knockdown of *E(z)* resulted in F1 sterility in the mini screen. This suggests that knockdown efficiencies were not optimal and/or a redundant role of the homologues *esc* and *esc1* and resulted in different effects on TE de-repression; different TE families appear to have different sensitivity to low levels of PRC2.

- Add statistics to Figure S2e and other panels where possible.

We added the statistical significance as calculated via Student’s t-test. In the revised version **Extended Data Fig. 2f** corresponds to the original **Fig. S2e**.

- Figure 4i is very difficult to read as there are so many colors and dots. Consider a more accessible panel layout.

We have edited the figure as requested and now only show half of the samples with their trajectories in the knockdown shown as arrow(head)s.

Julius Brennecke

Reviewer #2:

Remarks to the Author:

Rhi binding to H3K9me3-marked sites to turn on piRNA expression, which subsequently targets TE for degradation or silencing. The recruitment of Rhi to target sites is only partially explained by Kipf. In the manuscript, the authors first reported that the knockdown of Ez and several other PRC2 subunits in germ cells resulted in the derepression of transposons, suggesting a critical role of PRC2 and H3K27me3 on TE silencing through a partial contribution to Rhi function. They showed the Ez knockdown reduced 42AB piRNA, but not 20A and 80F clusters. Genome-wide analyses showed that H3K27me3 co-occupied with H3K9me3 and Rhi at 42AB loci. Overall, almost all Rhi-bound sites are also bound by H3K9me3, about half of which also contain H3K27me3 mark. The authors further showed that Rhi binding to some piRNA source loci is dependent on Ez, through Ez-knockdown assays. However, control experiments were not included to show that they are indeed independent of Kipf. Some of the key findings were further confirmed using S2 cells with the expression of synthetic dual-chromatin reader domains.

The manuscript is well-written, and the results are clearly described. The quality of the data presented is great. The findings that Ez is required for some piRNA function and TE expression and the presence of dual H3K9me3 and H3K27me3 at some piRNA source loci are novel and interesting. However, the experiments are limited in scope. Some essential control experiments that would have been very helpful in discerning the two mechanisms of Rhi recruitment to target loci were not comparatively done.

We thank the reviewer for their comments and valuable suggestions, which helped us to greatly improve the manuscript.

As such, the mechanism remains elusive. It seems H3K9me3 is also present at Ez-dependent Rhi binding sites, but their role is unclear.

Thank you for this valuable comment, which highlights the fact that we had not been completely explicit in the manuscript and that further clarification was needed. Rhino belongs to the HP1 family of chromatin regulators, which bind to H3K9me2/3 via their chromodomain (CD). Previous studies have shown that the chromodomain of Rhi resembles that of the canonical HP1 family member Su(var)205, specifically recognizing methylated H3K9 peptides but not unmodified H3 peptides (Mohn et al. 2014). In addition, a mutation in the H3K9 methyltransferase Eggless/SETDB1 has been shown to disrupt the transcription of piRNA source loci (Rangan et al. 2011). We added this information in the introduction.

Our results confirmed these findings, as we observed a complete loss of Rhino protein in germ cells upon *eggless* knockdown. This confirmation of previously reported dependency on Eggless has been added to the result section (**Extended Data Fig. 2a**). However, our study goes beyond the role of H3K9me3 and shows that Kipf-independent piRNA source loci additionally require the H3K27me3 methyltransferase E(z) for piRNA production.

Specific comments:

Fig.1: is H3K27me3 required for this role of Ez? Can an Ez methylation mutant rescue the phenotype?

That is a very relevant question. Joshi and colleagues have reported mutant alleles (E(z)1, E(z)son1, E(z)son3 and E(z)62 of the SET domain residues required for histone methyltransferase activity of E(z) (Joshi et al. 2008). However, as reported in this article, all four SET mutants display recessive lethality making them unsuitable for use in our study.

Nevertheless, we observe a clear decrease of H3K27me3 in E(z)-KD ovaries at Kipf-independent binding sites of Rhino, but not at Kipf-dependent sites (**Fig. 3g**). This strongly suggests that H3K27me3 contributes to Rhi recruitment, unless there is an unknown role of E(z) beside its methyltransferase activity.

Fig.2c. Can the authors show genome-wide distribution and overlap between Ez and Kipf? This is necessary to support the key conclusion that there are distinct Kipf- and Ez-mediated Rhi bindings.

We thank the reviewer for the comment. Due to the unavailability of a suitable antibody for ChIP-seq of E(z) in *Drosophila*, we did not perform a genome-wide distribution analysis for E(z). However, since E(z) is the histone methyltransferase responsible for H3K27 trimethylation (Czermin et al. 2002), we

have added a Venn diagram showing the overlap between Kipferl and H3K27me3 ChIP-seq signals across 1kb genomic bins, as suggested by the reviewer. Although there is a weak enrichment (1.7-fold) in overlap between H3K27me3 and Kipf, the vast majority of H3K27me3 regions (76%) do not overlap with Kipferl regions. We have included this result in the new **Extended Data Fig. 3c**. This also confirms our previous observation that H3K27m3 appears to be absent from Kipf-dependent Rhi binding sites (**Fig. 3g**), but present at E(z)-dependent Rhi binding sites, suggesting two distinct modes of Rhi recruitment.

Extended Data Fig. 3c: Venn diagram showing 1 kb bins occupied by Kipf and/or H3K27me3 peaks.

Fig.3. To support the conclusion that Ez and Kipf may provide two distinct modes of Rhi recruitment to separate sites, can the authors show how the knockdown of Kipf or Ez, respectively, affect Rhi binding at piRNA source loci?

The production of piRNAs in germline cells of *Drosophila* ovaries depends exclusively on recruitment of Rhi (Mohn et al. 2014). Thus, piRNA production can serve as measurement of Rhi binding, and the analysis of piRNAs shown in **Fig. 3b** illustrates that Kipf and E(z) have complementary roles in piRNA production (and therefore Rhi recruitment). Indeed, the stronger a region depends on Kipf, the less it depends on E(z) and vice versa. The referee's question highlighted that the connection between Rhi recruitment and piRNA production was not sufficiently clear and we have edited the manuscript to better emphasise this.

Nevertheless, we also agree that directly demonstrating the loss of Rhi recruitment following *Kipf* and *E(z)* knockdown provides an important additional control. Therefore, we have included this analysis, based on Rhi ChIP-seq, in a new panel (**Fig. 3d**; the previous panel is now **Extended Data Fig. 4a**). We again observe that the stronger Rhi binding depends on Kipf, the less it depends on E(z), and vice versa, suggesting two alternative modes of recruitment.

Moreover, the effect of *Kipf* knockdown on Rhino binding has already been documented by Baumgartner and colleagues (Baumgartner et al. 2022) (**Fig. 3e**). Their study showed that Rhino binding is affected at the *80F* piRNA cluster but not at *42AB*, which we identified as E(z)-dependent. Our findings collectively support the conclusion that E(z) and Kipferl provide distinct modes of Rhino recruitment to separate piRNA source loci.

Also, will they affect H3K9me3 and/or H3K27me3 binding at these sites? Presumably, Ez depletion will surely affect H3K27me3.

Please see below.

Fig.3e: what's the level of H3K9me3 and H3K27me3 on these sites in control and Ez-knockdown cells?

These two question are difficult to investigate rigorously as we performed H3K9me3 and H3K27me3 ChIP-seq on total ovaries (a mixed tissue consisting of numerous cell types and stages), in which E(z) expression is only abolished in the germline (using the *nos-Gal4* driver). However, **Extended Data Fig. 2b** supports our statement, showing a loss of H3K27me3 in the germ cells upon *E(z)* KD, whereas the staining remains unchanged in the somatic cells.

To address the reviewer's concerns, and to examine if the depletion of E(z) affects the levels of H3K27me3 and H3K9me3 at the piRNA source loci, we have nevertheless analysed the enrichment of both marks at these loci. Our ChIP-seq showed a decrease in H3K27me3 enrichment at the *42AB* piRNA cluster which is consistent with the loss of Rhi binding and piRNA production from this locus. However, the level of H3K27me3 at the Kipf-dependent piRNA source locus *80F* was not affected, consistent with low H3K27me3 levels on the locus. This new analysis, presented in **Extended Data**

Fig. 4d supports our results already shown in **Fig. 3g**, which demonstrates a clear decrease in H3K27me3 levels upon *E(z)* KD at the Kipf-independent Rhino peaks.

As highlighted in our manuscript, we noticed that our H3K9me3 antibody appeared to capture some H3K27me3 signal at regions with very strong H3K27me3 enrichment, as also stated previously in **Supplementary Note 2**. Therefore, we believe it is problematic to perform the same type of analysis with H3K9me3 ChIP-seq data. In addition to the genome browser view of the region covering the *sox102F* piRNA source locus (**Extended Data Fig. 4b**), we have also included a genome browser view of the *42AB* piRNA cluster (**Extended Data Fig. 4c**). These representations show that there is no significant loss of H3K9me3 at these loci upon *E(z)* knockdown.

Is the reduction of Rhi binding at 42AB caused by H3K27me3 loss in the knockdown cells?

We agree that we are not demonstrating that the reduction of Rhi binding at the *42AB* piRNA cluster is caused by H3K27me3 loss, what we observe is a correlation. As illustrated in the **Fig. 3g**, we observed a decrease of H3K27me3 levels upon *E(z)* KD on the Kipf-independent Rhi peaks which is correlated to a loss of Rhi binding to these loci, consistent with a role of H3K27me3 in Rhi recruitment. We have added a sentence to highlight this correlation: *“This suggests a strong correlation between the reduction in Rhi binding at Kipf-independent sites and the loss of H3K27me3 following E(z) knockdown.”*

Fig.3g: the information provided here is fragmented. Can the authors show H3K9me3 and H3K27me3 in control and kipf-knockdown cells in comparison to control and Ez-knockdown cells?

We thank the reviewer for this comment. H3K9me3 enrichment was previously measured in control and *Kipf* knockdown cells by Baumgartner and colleagues (Figure 3g in Baumgartner et al. 2022), and the authors reported no difference. Therefore, in **Fig. 3g** of our work, we specifically focused on comparing H3K27me3 and Rhi levels in control and *E(z)* or *Kipf* knockdown cells at Kipf-dependent and Kipf-independent Rhino peaks. This approach allows us to examine the specific effects of *E(z)* or *kipf* knockdown on H3K27me3 enrichment and Rhi binding at Kipferl-dependent and independent Rhino peaks and to control for background noise in ChIP-seq experiments. As requested, we have now added H3K27me3 in *Kipf* knockdown to **Fig. 3g**, illustrating that *Kipf* does not affect H3K27me3 levels.

Fig.4: similar to comments in Fig.3 above, the experimental design missed important controls to allow full comparison of the two categories of Ez- or kipf-dependent sites. For instance, 4c should include k27, 4d should include k9, and so forth.

The referee is referring to the Venn diagrams used in **Fig. 4c,d,e,h** to show peak overlaps across three histone marks (H3K9me3, H3K27me3, and H2AK118ub) and four chromodomain constructs (CD^{2xCBX1} , CD^{2xRhi} , CD^{2xPc} , $CD^{Pc-CBX1}$). We initially divided the overlap analyses into multiple figures in order not to overwhelm the reader with all information at the same time, and to keep the figures accurate.

As requested, we have added H3K27me3 to **Fig. 4c**, confirming that CD^{2xCBX1} recognizes H3K9me3, but not H3K27me3. Similarly, we have added H3K9me3 to the new **Extended Data Fig. 6b** (as we were unable to add a fifth group to **Fig. 4d**), confirming that CD^{2xPc} recognizes H3K27me3, but not H3K9me3. Moreover, we have included an UpSet plot (**Extended Data Fig. 6f**) to show all major overlaps between any combination of H3K9me3, H3K27me3, CD^{2xCBX1} , CD^{2xRhi} , CD^{2xPc} , and $CD^{Pc-CBX1}$.

In addition to these updates, the source data file provided contains all data underlying the overlap analysis and can thus be used to visualise any other combination.

Fig.5b: There is a strong correlation between H3K9me3 level, Rhi ChIP-seq signal, and piRNA expression. In contrast, the H3K27me3 level does not seem to be correlated with any of the above. In the site with low H3K9me3 (left of the curves), there were some bins with high H3K27me3. However, Rhi binding and piRNA expression were not increased in these bins.

We appreciate the reviewer's comment. Throughout the manuscript, we do not claim that H3K27me3 alone is sufficient for Rhi recruitment. Instead, bins with high H3K27me3 and low H3K9me3 likely correspond to canonical facultative heterochromatin, which is involved in gene regulation and make up a substantial part of the *Drosophila* genome. These regions are not normally piRNA source loci.

References

- Baumgartner L, Handler D, Platzer SW, Yu C, Duchek P, Brennecke J. 2022. The Drosophila ZAD zinc finger protein Kipferl guides Rhino to piRNA clusters. *Elife* **11**.
- Baumgartner L, Ipsaro JJ, Hohmann U, Handler D, Schleiffer A, Duchek P, Brennecke J. 2024. Evolutionary adaptation of an HP1-protein chromodomain integrates chromatin and DNA sequence signals. *Elife* **13**.
- Czermin B, Melfi R, McCabe D, Seitz V, Imhof A, Pirrotta V. 2002. Drosophila enhancer of Zeste/ESC complexes have a histone H3 methyltransferase activity that marks chromosomal Polycomb sites. *Cell* **111**: 185-196.
- Joshi P, Carrington EA, Wang L, Ketel CS, Miller EL, Jones RS, Simon JA. 2008. Dominant alleles identify SET domain residues required for histone methyltransferase of Polycomb repressive complex 2. *J Biol Chem* **283**: 27757-27766.
- Mohn F, Sienski G, Handler D, Brennecke J. 2014. The rhino-deadlock-cutoff complex licenses noncanonical transcription of dual-strand piRNA clusters in Drosophila. *Cell* **157**: 1364-1379.
- Rangan P, Malone CD, Navarro C, Newbold SP, Hayes PS, Sachidanandam R, Hannon GJ, Lehmann R. 2011. piRNA production requires heterochromatin formation in Drosophila. *Curr Biol* **21**: 1373-1379.
- Villasenor R, Pfaendler R, Ambrosi C, Butz S, Giuliani S, Bryan E, Sheahan TW, Gable AL, Schmolka N, Manzo M et al. 2020. ChromID identifies the protein interactome at chromatin marks. *Nat Biotechnol* **38**: 728-736.

Response to referees

Reviewer #1:

Remarks to the Author:

I apologise for the delay in reviewing the revised manuscript. The authors have, in my opinion, responded adequately to the points raised by both reviewers. The manuscript and the data presented are of high quality, the text and discussion are balanced, outline the new findings in the context of the field and point out the limitations of the work presented. Overall, this is an important paper that will stimulate further research in this exciting area of chromatin biology.

We thank the reviewer for their supportive comments and for recognizing the quality and significance of our work.

Reviewer #2:

Remarks to the Author:

The authors have addressed all of my concerns.

We are grateful to the reviewer for their valuable suggestions, which have helped improve our manuscript.